# Synthesis, Docking Studies and Pharmacological Evaluation of Serotoninergic Ligands Containing a 5-Norbornene-2-Carboxamide Nucleus

**DOI:** 10.3390/molecules27196492

**Published:** 2022-10-01

**Authors:** Rosa Sparaco, Ewa Kędzierska, Agnieszka A. Kaczor, Anna Bielenica, Elisa Magli, Beatrice Severino, Angela Corvino, Ewa Gibuła-Tarłowska, Jolanta H. Kotlińska, Giorgia Andreozzi, Paolo Luciano, Elisa Perissutti, Francesco Frecentese, Marcello Casertano, Anna Leśniak, Magdalena Bujalska-Zadrożny, Małgorzata Oziębło, Raffaele Capasso, Vincenzo Santagada, Giuseppe Caliendo, Ferdinando Fiorino

**Affiliations:** 1Dipartimento di Farmacia, Università Degli Studi di Napoli Federico II, Via D. Montesano, 80131 Napoli, Italy; 2Department of Pharmacology and Pharmacodynamics, Faculty of Pharmacy with Division of Medical Analytics, Medical University of Lublin, 4A Chodzki St., 20-093 Lublin, Poland; 3Department of Synthesis and Chemical Technology of Pharmaceutical Substances with Computer Modeling Laboratory, Faculty of Pharmacy, Medical University of Lublin, 4A Chodzki St., 20-093 Lublin, Poland; 4School of Pharmacy, University of Eastern Finland, Yliopistonranta 1, P.O. Box 1627, 70211 Kuopio, Finland; 5Chair and Department of Biochemistry, Medical University of Warsaw, Banacha 1 Str., 02-097 Warsaw, Poland; 6Dipartimento di Sanità Pubblica, Università Degli Studi di Napoli Federico II, Via Pansini, 5, 80131 Napoli, Italy; 7Centre for Preclinical Research and Technology, Department of Pharmacodynamics, Faculty of Pharmacy, Medical University of Warsaw, 1b Banacha Str., 02-097 Warsaw, Poland; 8Dipartimento di Agraria, Università di Napoli Federico II Via Università, 80055 Portici, Italy

**Keywords:** 5-norbornene-2-carboxilic acid, serotonin, arylpiperazine derivatives, 5-HT_1A_, 5-HT_2A_ and 5-HT_2C_ receptor ligands

## Abstract

A new series of 5-norbornene-2-carboxamide derivatives was prepared and their affinities to the 5-HT_1A_, 5-HT_2A_, and 5-HT_2C_ receptors were evaluated and compared to a previously synthesized series of derivatives characterized by exo-N-hydroxy-5-norbornene-2,3-dicarboximidenucleus, in order to identify selective ligands for the above-mentioned subtype receptors. Arylpiperazines represents one of the most important classes of 5-HT_1A_R ligands, and recent research concerning new derivatives has been focused on the modification of one or more portions of such pharmacophore. The combination of structural elements (heterocyclic nucleus, propyl chain and 4-substituted piperazine), known to be critical to the affinity to 5-HT_1A_ receptors, and the proper selection of substituents led to compounds with high specificity and affinity towards serotoninergic receptors. The most active compounds were selected for further in vivo assays to determine their functional activity. Finally, to rationalize the obtained results, molecular docking studies were performed. The results of the pharmacological studies showed that **Norbo-4** and **Norbo-18** were the most active and promising derivatives for the serotonin receptor considered in this study.

## 1. Introduction

Serotonin (5-hydroxytryptamine, 5-HT) is one of the major central neurotransmitters involved in many neuropsychiatric and neurological disorders, including depression, schizophrenia, migraine, pain, and Parkinson’s disease, to cite a few and, not surprisingly, its receptors have attracted interest as targets for therapeutic intervention. Early studies identified two subtypes of serotonin receptors (denoted S1 and S2) using radiolabeled ligand binding, but molecular biology and gene cloning techniques revealed that there are 14 different subtypes of receptors, divided into different families, based on sequence homology, signaling, and pharmacological criteria [1]. While 5-HT_3_Rs are cation-permeable ion channels, all the others are G-protein-coupled receptors (GPCR_s_) and are classified as rhodopsin-like receptors. From a therapeutic perspective, serotonin receptors (5-HTRs) represent an important challenge to identify novel drugs because of the numerous biological effects of the endogenous ligand. Serotonin plays a role in several physiological and pathological processes, including circadian rhythms, sexual and feeding behavior, thermoregulation, and cardiovascular function [2,3,4,5,6]. In addition, it is already known that 5-HT acts as a trophic, mitogenic, and anti-apoptotic factor for a wide range of normal and tumor cells [7,8]. One of the most prominent subtypes involved in serotonergic function is the 5-HT_1A_ receptor, which acts as a presynaptic autoreceptor to exert a powerful inhibitory control on serotonergic neurons in Raphe nuclei. It is also expressed as a postsynaptic heteroreceptor in multiple other brain regions, such as the hippocampus, amygdala, septum, hypothalamus, basal ganglia, and brain stem, associated with different neurological disorders, as well as known to be implicated in the proliferation of human tumor cells. 5-HT_1A_R antagonists inhibit the growth of different prostatic tumor cell lines, such as PC-3, DU-145, and LNCaP, as well as the proliferation of PC-3 xenografted subcutaneously in athymic nude mice [9]. Concerning the 5-HT_2_ receptor family (5-HT_2A_, 5-HT_2B_, and 5-HT_2C_), 5-HT_2_ARs activation presents promising neurotherapeutic targets. However, extensive structural homology complicates the design of selective agents. For example, 5-HT_2_-type receptors share 60–70% amino acid identity, and 27–31% identity with histamine H_1_ receptors within structurally conserved regions. Antagonism of 5-HT_2A_ receptors is associated with the improved efficacy of so-called atypical antipsychotics to treat schizophrenia and hallucinations. Additionally, inverse agonism of 5-HT_2C_ receptors, represents a tool for atypical antipsychotic poly-pharmacology and could be useful for the treatment of anxiety, major depression, and schizophrenia [10]. Arylpiperazines represent one of the most important classes of molecules approved for the management of neurological disorders. Mechanistically, almost all of them act as an agonist (partial/full) of the serotonin receptor (5-HT_1A_). Interestingly, close inspection of their structural framework reveals that most piperazine-based FDA-approved antidepressants demonstrate the presence of piperazine as a bridge between the two aromatic/heteroaromatic flanks. A general binding conformation of different FDA-approved antidepressants in the binding pocket of 5-HT_1A_ presents certain common features, including a salt-bridge with Asp-116, CH–π/π–π interactions, side chains of Phe361, and π–π stacking interactions with Phe-362. However, certain interactions are peculiar to specific molecules, including hydrogen bond interactions with Cys-187, Ser-199, and Asn-386. Further, detailed analysis of their bound conformation focusing on piperazine in the binding domain of serotonin receptor disclosed that the nitrogen atom of piperazine protonates at the physiological pH and forms a key salt bridge interaction with Asp116 of the binding domain of 5-HT_1A_. Along with other key interactions, this crucial contact stabilizes the complex of these therapeutic agents with the target. It also improves the binding affinity between the serotonin receptor and the piperazine bridge containing molecules. This key interaction also explains why piperazine moiety, a conventionally terminally attached moiety, is placed in the center of these molecules [11]. Anyway, a strong limitation in the potential use of many 5-HT_1A_ receptor ligands as pharmacological tools is represented by their undesired high affinity for other receptors. The dopaminergic D_2_ receptor and α_1_-adrenoceptor are two other examples of receptors for which several 5-HT_1A_ ligands show high affinity. Nevertheless, polypharmacology is considered an appropriate solution to achieve high efficacy complex therapy for neurological disorders. In fact, recent studies focusing on the search for new therapeutic agents indicates the importance of serotonin and dopamine receptors. However, it has already been demonstrated that dual- and multitarget acting compounds are useful against disorders affecting the central nervous system (CNS), which involve serotoninergic receptors and other GPCRs, such as muscarinic M4 receptors [12].

In our laboratories, there has been ongoing research to develop more selective serotoninergic ligands [13,14,15,16,17,18,19,20,21,22,23,24] to obtain novel pharmacological tools that could improve our knowledge of the signal transduction mechanism, leading to compounds with high affinity and selectivity. Previously described studies focused on the synthesis and pharmacological evaluation of a set of arylpiperazine derivatives containing an exo-N-hydroxy-5-norbornene-2,3-dicarboximide fragment nucleus. The binding data reported in these studies identified this original scaffold as an optimal structural element to enhance 5-HT_1A_ receptor affinity [14,16].

In continuation of our research program, we designed a new set of derivatives where the piperazine-*N*-alkyl moiety has been linked via propyl spacing unit to a 5-norbornene-2-carboxilic acid fragment as terminal part of long-chain arylpiperazines (LCAPs) (Figure 1); this choice was made considering this scaffold as a molecular simplification of derivatives previously synthesized, embodying the exo-*N*-oxy-5-norbornene-2,3-dicarboximide nucleus [14,16]. Consequently, we decided to investigate how the substitution of dicarboximide moiety presents in the exo-*N*-oxy-5-norbornene-2,3-dicarboximide nucleus with a simple amide bond, which may influence the binding affinity/selectivity profile. The 5-norbornene-2-carboxilic acid scaffold was linked via three methylene spacing unit to the *N*-4-aryl-substituted piperazines and this choice was done based on our previous investigations, where, as a general trend, compounds with piperazinyl propyl chain linked to different nuclei showed good and preferential affinity for the 5-HT_1A_R, with respect to compounds in which the spacer is one atom shorter. Finally, the use of 5-norbornene-2-carboxilic acid as a mixture of *endo/exo* allowed us to investigate the influence of *endo* and *exo* stereoisomerism of synthetized compounds on the binding affinity/selectivity profile towards the serotoninergic receptors investigated. All the new compounds were tested for their functional activity or affinity to 5-HT_1A_, 5-HT_2A_, and 5-HT_2C_ receptors and their multireceptor profiles were also evaluated in terms of functional activity for dopaminergic (D_2_). Moreover, compounds showing the best affinity and selectivity binding profiles towards serotoninergic receptors, have been evaluated by in vivo assay, i.e., behavioral tests, with the aim to discover novel pharmacological tools useful in treating psychiatric and neurological disorders, such as schizophrenia, depression and anxiety. Therefore, we evaluated the antipsychotic activity of the compounds in an amphetamine-induced hyperactivity test, antidepressant-like activity in the forced swim test (FST), and anxiolytic-like effects in the elevated plus-maze test (EPM). Moreover, we used additional tests, namely the spontaneous locomotor activity, rota-rod, and chimney tests, to assess potential adverse effects of the compounds. Results obtained from the tested compounds in the FST were compared with the commonly known antidepressant sertraline, and those in EPM with the clinically useful anxiolytic buspirone.

## 2. Results and Discussion

### 2.1. Chemistry

The general strategy for the synthesis of the target compounds (Table 1) is summarized in Figure 1. Briefly, 5-norbornene-2-carboxilic acid reacted with 3-chloropropan-1-amine hydrochloride in acetonitrile in the presence of *N*,*N*’-dicyclohexylcarbodiimide (DCC), hydroxybenzotriazole (HOBt) and triethylamine (TEA) to give the corresponding N-(3-chloropropyl)bicyclo [2.2.1]hept-5-ene-2-carboxamide (**2**). Subsequent condensation of intermediate (**2**) with the appropriate 4-X-substituted-piperazine performed in acetonitrile (CH_3_CN) with potassium carbonate (K_2_CO_3_) and sodium iodide (NaI), under reflux, provided the corresponding compounds as a mixture of endo and exo isomers. The purification of the reaction mixture and the separation of the isomers were carried out by silica gel open chromatography and further crystallization from the appropriate solvent. All new compounds were characterized by mono and bidimensional NMR spectroscopy and mass spectrometry. The endo and exo stereoisomerism of the two compounds was determined by comparing their spectroscopic data with those reported in the literature for related compounds. ^1^H-NMR, ^13^C-NMR, and MS for all final compounds were consistent with the proposed structures.

The olefinic protons of the *endo* compound of the norbornene nucleus resonate at chemical shift δ 5.97 and 6.21 while those of the exo compound resonate at δ 6.05 and 6.11. The comparison of these data with those presents in the literature allowed us to assign the *endo* and *exo* configuration to the two compounds. A further confirmation was possible thanks to the analysis of the ROESY spectrum. In the ROESY spectrum of the *endo* compound, there was a key correlation between the methine proton at δ 2.84 with the methylene bridge proton at δ 1.27, while in the exo compound there was a correlation between the methine proton at δ 1.96 with the olefin proton at δ 6.05 (Figure 1). These correlations together with the comparison of the data present in the literature allowed us to univocally assign the *endo* and *exo* stereoisomerism of the two compounds under examination.

### 2.2. Functional Activation of the 5-HT_1A_ and D_2_ Receptors

Potencies (EC_50_) and efficacies (E_max_) of 5-HT_1A_ receptor activation by a series of **Norbo-1-28** compounds are given in Table 1. As evidenced by the [^35^S]GTPγS functional assay, most compounds screened were agonists for the 5-HT_1A_ receptor with potencies in the submicromolar and micromolar range. Compounds’ potencies differed depending on the type of substitution on the phenyl ring but did not rely too much on their stereochemistry. As revealed by one-way ANOVA, substitution of the piperazine fragment with 2,3-dimethyl- (**Norbo-3**, **Norbo-4**), 4-methoxy- (**Norbo-7**, **Norbo-8**), 2-chloro- (**Norbo-9**, **Norbo-10**) or 3-CF_3_-phenyl (**Norbo-19**, **Norbo-20**) moieties, enhanced the potency of 5-HT_1A_ receptor activation by almost 10-fold for the *endo* isomers and 6-fold for the *exo* isomers on average. The EC_50_ values for the above-mentioned *endo* isomers were in the submicromolar range, i.e., 181 nM, 194 nM, 165 nM, 198 nM, and 195 nM for **Norbo-4**, **Norbo-8**, **Norbo-10,** and **Norbo-20**, respectively, while the calculated EC_50_ for the unsubstituted compound (**Norbo-2**) was 1633 nM. Potencies for the *exo* analogs ranged from 177.4 nM for **Norbo-3** to 347.8 nM for **Norbo-7**, while the potency of the mother **Norbo-1** was 1200 nM. 

Most of the *exo* and *endo* isomer pairs did not differ in terms of potency with only two exceptions. Namely, only the *endo* isomer was active among 4-chlorophenyl derivatives (**Norbo-11** and **Norbo-12**) (*p* < 0.001). On the other hand, the *exo* isomer showed greater potency among the 1,3-benzodioxole-derived compounds (**Norbo-25** and **Norbo-26**) (*p* < 0.05). 

In general, none of the substitutions introduced into the 4-phenylpiperazin-1-yl ring significantly increased 5-HT_1A_ activation efficacy. However, most of the compounds still expressed higher 5-HT_1A_ activation efficacies than the reference compound, 8-hydroxy-di-propylaminotetralin (8-OH-DPAT), classifying them as full agonists. However, the introduction of several substitutions proved detrimental for 5-HT_1A_ activation efficacy. Namely, derivatives bearing the 2-methoxyphenyl (**Norbo-5**, **Norbo-6**), 4-chlorophenyl (**Norbo-11**, **Norbo-12**), 4-fluorophenyl (**Norbo-17**, **Norbo-18**), and 2-chlorophenyl (**Norbo-9**) piperazine substituents were less efficacious when compared with the unsubstituted mother compounds **Norbo-1** or **Norbo-2**, respectively. Differences in efficacy were also evidenced among some pairs of stereoisomers. The *endo* isomers with monohalogenophenyl functionalities at the piperazine ring (**Norbo-10**, **Norbo-12**, **Norbo-16** and **Norbo-18**) were characterized by higher 5-HT_1A_ efficacies than their corresponding *exo* analogs. 

The study also identified some inactive derivatives expressing no agonist activity for the 5-HT_1A_ receptor. Namely, within the chlorine substituted analogs, all substitutions of the phenyl ring other than one *ortho*- positioned chlorine atom (**Norbo-9**) produced analogs devoid of 5-HT_1A_ agonist activity. Additionally, derivatives with the furoyl (**Norbo-27**, **Norbo-28**) terminal fragment also exerted no agonist activity at the 5-HT_1A_ receptor.

Finally, none of the compounds tested expressed no agonistic activity for the D_2_ receptor (Table 2). When screened for their antagonistic properties, the compounds only showed submilimolar inhibitory potencies against dopamine-induced stimulation of the D_2_ receptor. 

### 2.3. 5-HT_2A_ and 5-HT_2C_ Receptor Binding

All of the new compounds were tested for their affinity at 5-HT_2A_ and 5-HT_2C_ receptors. Some of the new synthesized derivatives showed interesting affinity values in the nanomolar range towards 5-HT_2A_ receptors and lower affinities for 5-HT_2C_ receptors (Table 3). Except for the outstanding 5-HT_2A_ receptor affinity and selectivity of compound **Norbo-14** (*Ki* = 17.93 nM with *pKi* = 7.75 ± 0,07) and **Norbo-18** (*Ki* = 18.65 nM with *pKi* = 7.73 ± 0.11). Other interesting *Ki* values were those of compounds **Norbo-17** (*Ki* = 22.66 nM with *pKi* =7.64 ± 0.08), **Norbo-20** (*Ki* = 22.86 nM with *pKi* = 7.64 ± 0.10), **Norbo-3** (*Ki* = 23.49 nM with *pKi* = 7.63 ± 0.08), and **Norbo-13** (*Ki* = 28.81 nM with *pKi* = 7.54 ± 0.08). Moreover, compound **Norbo-13** showed an interesting mixed 5-HT_2A_/5-HT_2C_ profile with *Ki* values of 28.81/122 nM, whereas compounds **Norbo-10**, **Norbo-11,** and **Norbo-15** presented the most attractive 5-HT_2C_ affinity profile with a *Ki* value of 13, 31 and 31 nM and *pKi* values of 7.87 ± 0.35, 7.51 ± 0.29 and 7.50 ± 0.25, respectively. As compared to the reference 5-HT_2A_ receptor ligand, ketanserin (*pKi* = 8.27 ± 0.06), one can conclude that compounds **Norbo-14**, **Norbo-18**, **Norbo-17**, **Norbo-20,** and **Norbo-3** expressed satisfactory affinities to the 5-HT_2A_ receptor. Simultaneously, the 3,4-dichloro-, 4-fluoro, 3-trifluoromethyl- and 2,3-dimethyphenyl piperazine substituents had the strongest influence on 5-HT_2A_ receptor binding affinity. When comparing the evaluated series of arylpiperazine derivatives to 5-HT_2C_ receptor selective ligands, such as RS-102221 (*pKi* = 8.25 ± 0.12), one can point to norbornene derivatives supporting the 2-chloro, 4-chloro, and 2-fluorophenylpiperazine moieties as terminal group (**Norbo-10**, **Norbo-11,** and **Norbo-15**) are the most promising, with their pKi values below 40 nM. 

The difference in affinity observed between this new series of derivatives (**Norbo 1–28**) and the previously described series [14,16] characterized by analog exo-N-oxy-5-norbornene-2,3-dicarboximide nucleus linked via two or three methylene spacing unit to 4-substituted piperazines, demonstrated that a simple amide bond (**Norbo 1–28**) instead of a dicarboximide moiety represents a critical feature in determining differences in binding with 5-HTRs. 

In fact, regarding these novel derivatives, although they have a lower affinity profile than those previously synthesized, the influence of this original scaffold associated to the appropriate substituents on the phenylpiperazine ring and heterocyclic nucleus were particularly profitable to enhance 5-HT_2A_ and 5-HT_2C_ receptor affinity. Anyway, in order to rationalize the differential binding affinities/activities, molecular docking studies were carried out on the complete series of derivatives. 

### 2.4. In Vitro Evaluation of 5-HT-Evoked Contractions

Successively, the compounds **Norbo-5**, **Norbo14**, **Norbo-16**, **Norbo-17**, **Norbo-18**, **Norbo-19,** and **Norbo-20** with better affinity/selectivity binding profiles towards 5-HT_2A_ receptors were tested by in vitro assay to determine their activity on 5-HT-evoked contractions. In the rat ileum, 5-HT_2A_ receptors are located on smooth muscles and their activation by 5-HT is known to induce contraction. Consequently, 5-HT_2A_ antagonists depress 5-HT-induced contractions in the rat ileum [24]. According to Briejer and colleagues, we have shown that 5-HT contracted the rat ileum longitudinal muscle. In preliminary experiments we found that the neuronal blocker tetrodotoxin (0.3 µM), the muscarinic receptor antagonist atropine (1 µM), the adrenergic receptor antagonists phentolamine (10^−6^ M) plus propranolol (10^−6^ M) did not affect the contractions by 5-HT. In contrast, ketanserin (0.1 µM), at concentration that blocks 5-HT_2A_ receptors, depressed the contractions induced by 5-HT. Collectively, these results suggest that 5-HT contracts the ileum by acting on 5-HT_2A_ receptors located on smooth muscle while muscarinic or adrenergic receptors are not involved. Results show the potency (expressed by the IC_50_ value) and the efficacy (expressed by the E_max_ value) of the compounds under investigation in inhibiting 5-HT-induced contractions in the rat ileum (a pharmacological assay useful to detect activity towards 5-HT_2A_ receptors). The compounds under investigation did not significantly inhibit the contractions induced by 5-HT. The rank order of efficacy was: **Norbo-16** (3.51 × 10^−7^ M) > **Norbo-5** (7.48 × 10^−7^ M) > **Norbo-20** (7.51 × 10^−7^ M) > **Norbo14** (9.63 × 10^−7^ M) > **Norbo-18** (3.06 × 10^−6^ M) > **Norbo-17** (3.25 × 10^−6^ M) > **Norbo-19** (3.56 × 10^−6^ M)**.** Concerning the potency, these compounds displayed potency approximately in the 10^−7^–10^−6^ M range, specifically, the rank order of potency was **Norbo-19** (Emax = 39.00%) > **Norbo-17** (Emax = 32.50%) > **Norbo-18** (Emax = 25.28%) > **Norbo-16** (Emax = 23.47%) > **Norbo14** (Emax = 22.57%) > **Norbo-20** (Emax = 21.79%) > **Norbo-5** (Emax = 15.42%). Finally, none of the compounds under investigation contracted per se, the rat ileum.

### 2.5. Molecular Docking Studies

In order to study the ligand-receptor interactions at the molecular level and to rationalize the observed structure–activity relationships, compounds **Norbo 1–28** were docked to the orthosteric sites of 5-HT_1A_, 5-HT_2A_, and 5-HT_2C_ receptors. The visualization of molecular interactions of selected most potent compounds with the receptors under investigation is shown in Figure 1, Figure 2 and Figure 3. As the studied compounds follow the classical pharmacophore model for the aminergic G protein-coupled receptor (GPCRs) ligands [25], the electrostatic interaction between the protonatable nitrogen atom of the ligand and the conserved Asp 3.32 of the receptor is a key ligand-receptor contact for all studied complexes [26]. In the case of most ligand-receptor complexes, Trp 6.48 and Phe 6.52 are crucial residues engaged in π-π stacking contact with *N*-aryl moieties of the ligands, as found for many similar complexes [22,23,27,28,29]. These residues, accompanied by Phe 6.51 and His 6.55, constitute an aromatic microdomain of, i.a., serotonin and dopamine receptors [30]. The importance of these residues for binding ritanserin, an inverse agonist of serotonin 5-HT_2C_ receptor, was confirmed by X-ray crystallography (PDB ID: 6BQH [6]) and verified by the mutation of Phe 5.47, Phe 6.44, and Trp 6.48. In addition, the serotonin receptor subtype selectivity may be governed by residues from the second extracellular loop (ecl2) [31], which serves for recognition of the “message” part of the ligand [32].

The mode of binding of the studied ligands is more differentiated for 5-HT_1A_ (Figure 2) and similar for 5-HT_2A_ (Figure 3) and 5-HT_2C_ (Figure 4) receptors and receptors. In the case of most ligands interacting with 5-HT_2A_ and 5-HT_2C_ receptors, the ligands adopt an extended conformation parallel to the transmembrane helix bundle. The norbornene moiety of the ligands is directed towards the extracellular vestibule while the *N*-aryl group penetrates deeper into receptor aromatic microdomain which is in accordance with our previous results [22,23]. This is also in line with the data recently published by Kucwaj-Brysz et al. [33], who identified molecular docking poses of serotonin receptor ligands with the arylpiperazine moiety deeply buried in the binding pocket. Moreover, such a ligand binding mode is in agreement with the one found for ritanserin in 5-HT_2C_ receptor (PDB ID: 6BQH [6]). In the case of 5-HT_1A_ receptor, some ligands are able to adopt an extended conformation parallel to the transmembrane bundle similar to that described above (e.g., **Norbo-3** and **Norbo-4**, see Figure 2A and Figure 2B, respectively), some adopt an extended conformation, but they are situated at the angle of 30–60 degrees regarding the transmembrane bundle (e.g., **Norbo-8**, Figure 2C, **Norbo-10**, Figure 2E, and **Norbo-20**, Figure 2F), while others adopt a bent conformation with norbornene moiety, penetrating deeper in the receptor cavity than the N-aryl group, e.g., **Norbo-8** (Figure 2C).

The affinity of the studied ligands to the serotonin receptors is mainly governed by the type of N-aryl group, as previously reported by us [22,23] and by Zagórska et al. [34] for similar series. Many potent compounds bear a halogen substituent in the aryl ring which can be rationalized by the possibility of halogen bond formation as suggested by Partyka et al. [35]. The exo/endo isomerism seems to have a less important effect as it displays no clear trend and is case-specific. This can be explained by the fact that the norbornene moiety of the ligand directs in most complexes towards the extracellular part of the receptor, constituted by flexible loops which can accommodate both isomers. However, regarding the general pharmacological profile of the compounds, the endo isomers are more beneficial, and they have been selected for animal studies (**Norbo-4**, **Norbo-8**, **Norbo-10**, **Norbo-14**, **Norbo-18**, and **Norbo-20**). The selectivity of the studied ligands to the serotonin receptor subtypes is first governed by the residues from the ecl2. Moreover, in case of 5-HT_1A_ receptor Phe 3.28, Ser 5.43, and Tyr 7.42 were found to be important in many ligand-receptor complexes. In summary, the performed molecular modeling study can facilitate the design of subsequent compound series with the expected serotonin receptor subtype selectivity or polypharmacology.

### 2.6. In Vivo Behavioral Test

Mental disorders represent a specific group of diseases since their course is miscellaneous in different patients. In fact, each patient should be treated individually, and drugs should possess multidirectional action, which increases the chances of amending the health. There is a great need to improve the treatment and dose selection for patients with mental disorders. Hence, the search for new drugs and effective therapeutic solutions is a constant problem.

Behavioral pharmacology research is the cornerstone of understanding the processes underlying the behavior of living organisms, as well as the biological basis of the behavioral, emotional and cognitive disorders that affect humans. Discoveries in this area have helped to explore the potential therapeutic effects of many substances in treating these disorders. Since biochemical abnormalities causing psychiatric disorders are not limited to single signaling pathways, a variety of screening tests are used in experimental pharmacology to reveal new potential drugs. Basic research in laboratory animals is a promising approach to study behavioral abnormalities associated with mental disorders and to identify new pharmacological therapies to address mental health challenges. Knowledge of pharmacology allows us to understand that chemicals with very specific structures and properties that, at controlled doses, can interact with the normal physiology processes to produce health-enhancing effects known as therapeutic effects. However, if dosages are insufficient or excessive, the effects will be useless or harmful (toxic), respectively [36].

Compounds **Norbo-4**, **Norbo-8**, **Norbo-10**, **Norbo-14**, **Norbo-18**, and **Norbo-20** were selected for further functional in vivo studies. The first part of experiments included: motor coordination and locomotor activity tests, generally accepted as basic procedures in central activity investigations of new agents [37]. Firstly, all compounds were administered at the dose of 30 mg/kg. It was observed that three of them (**Norbo-4**, **Norbo-14** and **Norbo-18**) disturbed the behavior of mice in the chimney test (Figure 5A) and impaired motor coordination assessed in the rota-rod test (Figure 5B). In the locomotor activity test, the same compounds at dose of 30 mg/kg decreased spontaneous motility after 6 and 20 min (Figure 6) of observation. Similarly, **Norbo-4** induced the same effect at the dose of 15 mg/kg. In the second stage of this study, we evaluated the antipsychotic ability of the new compounds. Presently, animal models of schizophrenia, commonly employed for preclinical studies of antipsychotic properties of drugs, regard mainly amphetamine and MK-801 models [38]. The first model is based on the manipulation of the dopaminergic system activity, and it may primarily respond to drugs that affect this neurotransmitter system. Many neuroleptics acting as dopaminergic antagonists reverse this effect [38]. Noteworthy, the hyperlocomotion following amphetamine is also sensitive to other classes of drugs [39]. On the other hand, several preclinical tests have pointed to the role of 5-HT_2C_ ligands in the modulation of monoaminergic systems, including dopaminergic. Indeed, dysfunction in serotoninergic activity could contribute to the alteration of dopaminergic function seen in schizophrenia [40]. Nevertheless, in the amphetamine model, none of the tested compounds at the dose of 15 mg/kg reduced amphetamine-induced hyperactivity of mice (Figure 7). Furthermore, due to the modulation of the central serotonin neurotransmission, the new compounds may also show anxiolytic and/or antidepressant activity [41,42]. Considering this premise, as well as in vitro data obtained for the tested compounds (mixed 5-HT_1A_/5-HT_2C_ affinity profile for all the compounds), we examined their antidepressant and anxiolytic potential in behavioral models commonly used in mice, i.e., FST and EPM. 

FST is a simple and fast test established in experimental pharmacology for detecting the antidepressant effect of tested substances [43]. The mouse placed in a beaker with water, without the possibility of escaping, initially makes intensive attempts to get out, but after a short time it gives up and adopts an attitude known as immobility. Selective serotonin reuptake inhibitors (SSRIs) are an example of antidepressants active in this test. They block the serotonin transporter (SERT) protein, which transports serotonin, and reduce its reuptake from the synaptic cleft. Due to their effectiveness, they are one of the most commonly used drugs in the treatment of depression. This group includes, among others, sertraline, which is a 5-HT_1A_ agonist. Sertraline at the dose of 15 mg/kg used in our experiments significantly shortened the immobility time of mice. The tested compound **Norbo-4** at three doses of 15, 7.5, and 3.75 mg/kg and **Norbo-18** (at the doses of 15 and 7.5 mg/kg) showed an antidepressant effect similar to fluoxetine, which was manifested by a significant reduction of immobility time of mice (Figure 8). This effect should be considered specific as the tested compound did not show a negative effect on the locomotor activity of mice (in the two lowest used doses) nor induce sedation (at a dose of 15 mg/kg). This suggests that the mice were able to cope with the sedation in this stressful confined space swimming situation. The twice-as-high dose was found to inhibit the CNS so strongly that the immobility time in this group was comparable to the control group. Meanwhile, the other tested compounds, at used doses, remained inactive in this test. Anxiety and stress-related disorders represent severe mental health conditions that affect the performance of daily tasks and represent a high cost to public health. Charles Darwin’s preliminary observation that animals and humans have similar traits in expressing emotions opens the possibility to study the mechanisms of mental disorders in other mammals (mainly rodents). The animal test for assessing anxiolytic effects, EPM, is based on the test animal’s aversion to open spaces [44]. Fear induced inhibition of exploratory activity affects entries into open arms in this task as a significant increase in the percentage of time spent on the open arms and the number of entries into open arms is observed with drugs that are clinically effective anxiolytics [45]. Therefore, this model has been used to assess the anxiolytic-like activity of new putative anxiolytic compounds [46]. Buspirone (5 mg/kg), used as a reference anxiolytic drug, also prolonged the time and increased the percentage of entries into the open arms (Figure 9). As in the FST test, also in this test, the **Norbo-4** compound, used in three lower doses, i.e., 15, 7.5, and 3.75 mg/kg, increased the percentage of open arm exploration time, analogous to buspirone. Additionally, the increase in the percentage of entries into the open arms was statistically significant. The remaining compounds at the used doses were inactive in this test.

## 3. Conclusions

We have described the synthesis of a new series of arylpiperazines as serotoninergic ligands (**Norbo 1–28**). The 4-methoxyphenyl and 2,3-dimethylphenyl piperazine derivatives supporting an 5-norbornene-2-carboxamide scaffold as terminal fragment (**Norbo-4** and **Norbo-8**) afforded a favorable agonistic profile for 5-HT_1A_ receptors (pEC_50_ = 6.74 ± 0.08 and pEC_50_ = 6.78 ± 0.09 respectively), whereas, besides the outstanding 5-HT_2A_ receptor affinity and selectivity of compounds **Norbo-14** (Ki = 17.93 nM) and **Norbo-18** (Ki = 18.65 nM), other interesting Ki values included that of compound **Norbo-20** (Ki = 22.86 nM), while compounds **Norbo-10** and **Norbo-11** presented the most attractive 5-HT_2C_ affinity profile with Ki values of 13 and 31 nM, respectively. 

Based on the in vitro results, the compounds **Norbo-4**, **Norbo-8**, **Norbo-10**, **Norbo-14**, **Norbo-18**, and **Norbo-20** were selected for further in vivo studies to investigate their functional activity. 

The obtained results showed that compounds **Norbo-4** and **Norbo-18** exerted antidepressant-like effects. Interestingly, the compound **Norbo-4** revealed significant anxiolytic properties, and in the EPM test it was almost as efficacious as buspirone. However, further pharmacological studies are necessary to determine detailed mechanism of action and clinical prospective usefulness of the new compounds.

In conclusion, data presented in this study confirm that, as obtained with the series previously synthesized [14,16], the novel synthesized compounds display a general trend of affinity towards serotoninergic receptors investigated. Molecular docking studies supported these results, highlighting some selective and additional interactions of the identified ligands with the investigated receptor subtype.

## 4. Materials and Methods

### 4.1. Synthesis

#### 4.1.1. General Procedures

All reagents substituted piperazines and solvents were commercial products obtained from Merck (Darmstadt, Germany). Melting points, determined using a Buchi Melting Point B-540 (Flawil, Switzerland) instrument, are uncorrected and represent values obtained on recrystallized or chromatographically purified material. ^1^H and ^13^C-NMR spectra were recorded on Bruker Advanced 400 MHz spectrometer (USA). COSY, HSQC and ROESY experiments were recorded on Bruker Advanced 700 MHz spectrometer (USA). Unless otherwise stated, all spectra were recorded in CDCl_3_. Chemical shifts are reported in ppm. The following abbreviations are used to describe peak patterns when appropriate: s (singlet), d (doublet), t (triplet), m (multiplet), q (quartet), qt (quintet), dd (doublet of doublet), td (triplet of doublets), and bs (broad singlet). Mass spectra of the final products were recorded on a LTQ-XL mass spectrometer equipped with an HESI ion source (Thermo Fisher Scientific, Waltham, MA, USA). Elemental analyses were carried out on a Carlo Erba model 1106. Analyses indicated by the symbols of the elements were within ±0.4% of the theoretical values. All reactions were followed by thin-layer chromatography, carried out on Merck silica gel 60 F_254_ plates with a fluorescent indicator, and the plates were visualized with UV light (254 nm). Preparative chromatographic purifications were performed using a silica gel column (Kieselgel 60). Solutions were dried over Na_2_SO_4_ and concentrated with a Buchi R-114 rotary evaporator at low pressure. 

#### 4.1.2. Synthesis of N-(3-Chloropropyl)bicyclo[2.2.1]hept-5-ene-2-carboxamide (**2**)

Commercially available 5-norbornene-2-carboxilic acid (mixture of *endo* and *exo*, predominantly *endo*) (1.00 g; 7.24 mmol) was solubilized in acetonitrile (10 mL) and cooled to 0 °C for 30 min. *N*,*N*’-dicyclohexylcarbodiimide (DCC, 1.64 g; 7.96 mmol) and hydroxybenzotriazole (HOBt, 1.22 g; 7.96 mmol) were added and the mixture was stirred for one hour. Finally, triethylamine (TEA, 1.11 mL; 7.96 mmol) and 3-chloropropylamine hydrochloride (0.94 g; 7.24 mmol) were added and the reaction mixture was stirred at room temperature for 8 h. When the reaction was completed, it was cooled to 0 °C. Subsequently, the mixture was filtered and evaporated under reduced pressure. The resulting residue was diluted with DCM (30 mL) and washed with NaHCO_3_ 5% and brine. The organic phase was dried over Na_2_SO_4_ and concentrated, yielding the desired intermediate ***2*** as a brown oil (yield: 70%), the mixture of *exo*/*endo* isomers of which was used without further purification in the next steps. ^1^H-NMR spectra for purified *exo*/*endo* intermediates were consistent with the proposed structures.

***exo*:**^1^H-NMR (400 MHz, CDCl3) δ:1.29 (d, 1H, J = 1.9 Hz), 1.31–1.36 (m, 1H), 1.68 (d, 1H, J = 8.2 Hz), 1.88–1.91 (m, 1H), 1.97–2.03 (m, 3H), 2.90 (s, 2H), 3.39 (q, 2H, -NH-CH2-, J = 6.4 Hz), 3.56 (t, 2H, J = 6.3 Hz), 6.09 (dd, 1H, J = 2.2 Hz), 6.13(dd,1H, J = 2.2 Hz). ^13^C-NMR (101 MHz, CDCl3) δ: 30.64, 32.31, 37.26, 41.27, 42.76, 44.81, 46.48, 47.32, 136.08, 138.39, 176.05; ESI-MS *m/z* [M+H]^+^ calculated for C_11_H_16_ClNO 213.70, Found = 214.4

***endo*:**^1^H-NMR (400 MHz, CDCl_3_) δ:1.29 (d, 1H, *J* = 3.3 Hz), 1.32–1.33 (m, 1H), 1.44 (dd, 1H, *J* = 8.2, 2.0 Hz), 1.91–1.99 (m, 3H), 2.84–2.88 (m, 1H), 2.92 (s, 1H), 3.13 (s, 1H), 3.33–3.37 (m, 2H, -NH-CH_2_-), 3.54 (t, 2H, *J* = 6.4 Hz), 5.95 (dd, 1H, *J* = 5.5, 2.7 Hz), 6.22 (dd, 1H, *J* = 5.5, 3.1 Hz). ^13^C-NMR (101 MHz, CDCl_3_) δ: 30.06, 32.21, 37.17, 42.82, 42.86, 44.99, 46.33, 50.20, 132.31, 138.04, 174.66; ESI-MS *m/z* [M+H]+ calculated for C_11_H_16_ClNO 213.70, Found = 214.5

#### 4.1.3. General Procedure for the Synthesis of Norbornene Derivatives (**Norbo-1-28**)

To a solution of N-(3-chloropropyl) bicyclo[2.2.1]hept-5-ene-2-carboxamide (**2**, 1.00 g; 4.70 mmol) in acetonitrile, sodium iodide (NaI, 0.77 g; 5.17 mmol) was added. The mixture was heated at reflux and stirred for 30 min. T

Then, the appropriate 4-X-substitued-piperazine (4.70 mmol) and potassium carbonate (K_2_CO_3_, 0.71 g; 5.17 mmol) were added. The reaction was stirred at reflux overnight. Subsequently, the mixture was filtered, and the solvent was removed in vacuo. The resulting product was diluted with DCM (30 mL) and washed with water and brine. The organic layer was dried over Na_2_SO_4_ and concentrated. The residue was purified by silica gel open chromatography using dichloromethane/methanol (9:1 *v/v*) as eluent to obtain the final compound as a mixture of *endo* and *eso* isomers. The separation of two isomers was carried out by silica gel open chromatography with diethyl ether/methanol (8:2 *v/v*) as eluent. The combined and evaporated product fractions were crystallized from diethyl ether or converted into the corresponding hydrochloride salt, yielding the desired products (**Norbo 1–28**) as white solids.

#### 4.1.4. Synthesis of Exo-N-(3-(4-phenylpiperazin-1-yl) propyl)bicyclo[2.2.1]hept-5-ene-2-carboxamide (**Norbo-1**) and endo-N-(3-(4-phenylpiperazin-1-yl)propyl)bicyclo[2.2.1]hept-5-ene-2-carboxamide (**Norbo-2**)

Following the synthetic procedure reported above, **Norbo-1** and **Norbo-2** were synthetized starting from **2** (1.00 g; 4.70 mmol) and 1-phenylpiperazine (0.76 g; 4.70 mmol).

**Norbo-1:** Yield: 33%; mp: 120.8–121.8 °C; ^1^H-NMR (400 MHz, CDCl_3_) δ: 1.27–1.35 (m, 2H), 1.68 (d, 1H, *J* = 8.5 Hz), 1.78–1.81 (m, 2H), 1.90–1.92 (m, 1H), 1.96–1.98 (m, 1H), 2.59 (t, 2H, -CH_2_-N^1^-, *J* = 6.0 Hz), 2.73 (bs, 4H, 2CH_2_ pip., *J* = 4.7 Hz), 2.89 (s, 1H), 2.93 (s, 1H), 3.27 (bs, 4H, 2CH_2_ pip., *J* = 4.5 Hz), 3.38–3.40 (m, 2H, -NH-CH_2_-), 6.07 (dd, 1H, *J* = 5.5, 3.0 Hz), 6.11(dd, 1H, *J* = 5.3, 3.1 Hz), 6.87–6.94 (m, 3H), 7.00 (bs, 1H, NH), 7.28–7.30 (m, 2H); ^13^C-NMR (101 MHz, CDCl_3_) δ: 25.00, 30.39, 39.25,41.54, 44.88, 46.35, 47.22, 48.98, 53.16, 57.34, 116.20, 120.16, 129.18, 135.96, 138.19, 150.84, 175.52; ESI-MS *m/z* [M+H]^+^ calculated for C_21_H_29_N_3_O 339.47, Found = 340.24; Anal. Calcd for C_21_H_29_N_3_O: C, 74.30; H, 8.61; N, 12.38. Found C, 74.52; H, 8.60; N, 12.40.

**Norbo-2:** Yield: 30%; mp: 113.7–114.1 °C; ^1^H-NMR (400 MHz, CDCl_3_) δ:1.26 (d, 1H, J = 8.2 Hz), 1.32–1.36 (m, 1H), 1.41 (dd, 1H, J = 8.2, 1.7 Hz), 1.66–1.73 (m, 3H), 1.87–1.94 (m, 1H), 2.47 (t, 2H, -CH_2_-N^1^-, J = 6.4 Hz), 2.62 (bs, 4H, 2CH_2_ pip., J = 4.7 Hz), 2.81–2.86 (m, 1H), 2.89 (s, 1H), 3.13 (s, 1H), 3.21 (bs, 4H, 2CH_2_ pip., J = 4.7 Hz) 3.29–3.34 (m, 2H, -NH-CH_2_-), 5.97 (dd, 1H, J = 5.5, 2.7 Hz), 6.20 (dd, 1H, J = 5.5, 3.0 Hz), 6.57 (bs, 1H, NH), 6.85 (t, 1H, J = 7.3 Hz), 6.93 (d, 2H, J = 8.2 Hz), 7.28 (d, 2H, J = 7.7 Hz); ^13^C-NMR (101 MHz, CDCl_3_) δ: 24.94, 30.39, 39.16, 41.54, 44.86, 46.35, 47.22, 48.91, 53.13, 57.26, 116.24, 120.21, 129.19, 135.96, 138.19, 150.80, 175.57; ESI-MS *m/z* [M+H]^+^ calculated for C_21_H_29_N_3_O 339.47, Found = 340.24; Anal. Calcd for C_21_H_29_N_3_O: C, 74.30; H, 8.61; N, 12.38. Found C, 74.07; H, 8.59; N, 12.42.

#### 4.1.5. Synthesis of Exo-N-(3-(4-(2,3-dimethylphenyl) piperazin-1-yl)propyl)bicyclo[2.2.1]hept-5-ene-2-carboxamide (**Norbo-3**) and endo-N-(3-(4-(2,3-dimethylphenyl) piperazin-1-yl)propyl)bicyclo[2.2.1]hept-5-ene-2-carboxamide (**Norbo-4**)

Following the synthetic procedure reported above, **Norbo-3** and **Norbo-4** were synthetized starting from **2** (1.00 g; 4.70 mmol) and 1-(2,3-dimethylphenyl) piperazine (0.89 g; 4.70 mmol).

**Norbo-3:** Yield: 32%; mp: 109.1–111.4 °C; ^1^H-NMR (400 MHz, CDCl_3_) δ:1.29–1.36 (m, 2H), 1.69 (d, 1H, *J* = 8.5 Hz), 1.77–1.80 (m, 2H), 1.92–1.95 (m, 1H), 2.01–2.03 (m, 1H), 2.21 (s, 3H), 2.27 (s, 3H), 2.62 (t, 2H, -CH_2_-N^1^-, *J* = 6.0 Hz), 2.73 (bs, 4H, 2CH_2_ pip., *J* = 4.7 Hz), 2.91–2.97 (m, 6H), 3.38–3.40 (m, 2H, -NH-CH_2_-), 6.10 (dd, 1H, *J* = 5.5, 3.2 Hz), 6.12 (dd, 1H, *J* = 5.5, 3.1 Hz), 6.89–6.93 (m, 2H), 7.07 (t, 1H, *J* = 7.0 Hz), 7.20 (bs, 1H, NH); ^13^C-NMR (101 MHz, CDCl_3_) δ: 13.89, 20.59, 24.85, 30.35, 39.41, 41.56, 44.93, 46.36, 47.25, 51.83, 53.68, 57.49, 116.55, 125.27, 125.90, 131.23, 135.98, 138.01, 138.19, 150.97, 175.49; ESI-MS *m/z* [M+H]^+^ calculated for C_23_H_33_N_3_O 367.53, Found = 368.27; Anal. Calcd for C_23_H_33_N_3_O: C, 75.16; H, 9.05; N, 11.43. Found C, 75.23; H, 9.06; N, 11.45.

**Norbo-4:** Yield: 38%; mp: 119.8–120.6 °C; ^1^H-NMR (400 MHz, CDCl_3_) δ:1.28 (d, 1H, J = 8.0 Hz), 1.35–1.40 (m, 1H), 1.43–1.45 (m, 1H), 1.73–1.76 (m, 2H), 1.90–1.95 (m, 1H), 2.22 (s, 3H), 2.27 (s, 3H), 2.56 (t, 2H, -CH_2_-N^1^-, *J* = 5.5 Hz), 2.71(bs, 4H, 2CH_2_ pip., *J* = 4.7 Hz), 2.86–2.88 (m, 1H), 2.91 (s, 1H), 2.97 (bs, 4H, 2CH_2_ pip., *J* = 4.7 Hz), 3.18 (s, 1H), 3.31 (t, 2H, -NH-CH_2_-, *J* = 5.1 Hz), 5.98 (dd, 1H, *J* = 5.4, 2.7 Hz), 6.20 (dd, 1H, *J* = 5.4, 3.1 Hz), 6.78 (bs, 1H, NH), 6.91 (d, 1H, *J* = 7.6 Hz), 7.07–7.11 (m, 2H) ^13^C-NMR (101 MHz, CDCl_3_) δ: 13.90, 20.59, 25.07, 29.79, 39.06, 42.67, 44.91, 46.15, 49.87, 51.79, 53.77, 57.36, 116.54, 125.23, 125.89, 131.25, 132.43, 137.51, 138.08, 151.06, 174.22; ESI-MS *m/z* [M+H]^+^ calculated for C_23_H_33_N_3_O 367.53, Found = 368.27; Anal. Calcd for C_23_H_33_N_3_O: C, 75.16; H, 9.05; N, 11.43. Found C, 75.00; H, 9.03; N, 11.39.

#### 4.1.6. Synthesis of Exo-N-(3-(4-(2-methoxyphenyl) piperazin-1-yl)propyl)bicyclo[2.2.1]hept-5-ene-2-carboxamide (**Norbo-5**) and endo-N-(3-(4-(2-methoxyphenyl) piperazin-1-yl)propyl)bicyclo[2.2.1]hept-5-ene-2-carboxamide (**Norbo-6**)

Following the synthetic procedure reported above, **Norbo-5** and **Norbo-6** were synthetized starting from **2** (1.00 g; 4.70 mmol) and 1-(2-methoxyphenyl) piperazine (0.90 g; 4.70 mmol). The final isomer **Norbo-5** was converted into the corresponding hydrochloride salt.

**Norbo-5**: Yield: 45%; mp: 194.8–195.7 °C; ^1^H-NMR (400 MHz, DMSO-*d*_6_) δ: 1.15 (m, 2H), 1.61 (d, 1H, *J* = 8.3 Hz), 1.75–1.78 (m, 2H), 1.84–1.87 (m, 1H), 2.02–2.03 (m, 1H), 2.81 (bs, 4H, 2CH_2_ pip., *J* = 4.5 Hz), 3.00–3.14 (m, 8H), 3.44 (m, 2H, -NH-CH_2_-), 3.77 (s, 3H, -OCH_3_), 6.10 (dd, 1H, *J* = 5.5, 2.6 Hz), 6.12(dd,1H, *J* = 5.5, 2.9 Hz), 6.86–6.93 (m, 2H), 6.95–7.00 (m, 2H), 8.06 (bs, 1H, NH); ^13^C-NMR (101 MHz, DMSO-*d*_6_) δ: 24.23, 30.28, 36.39, 40.60, 41.42, 43.49, 46.13, 47.28, 51.59, 53.96, 55.82, 112.39, 118.69, 121.29, 123.93, 136.67, 138.23, 139.78, 152.26, 175.34; ESI-MS *m/z* [M+H]^+^ calculated for C_22_H_31_N_3_O_2_ 369.50, Found = 370.0; Anal. Calcd for C_22_H_31_N_3_O_2_: C, 71.51; H, 8.46; N, 11.37. Found C, 71.65; H, 8.47; N, 11.40.

**Norbo-6:** Yield: 31%; mp: 83.9–86.1 °C; ^1^H-NMR (400 MHz, CDCl_3_) δ: 1.27 (d, 1H, *J* = 8.0 Hz), 1.34 (dt, 1H, *J* = 6.5, 2.1 Hz), 1.41 (d, 1H, J = 6.6 Hz), 1.79–1.82 (m, 1H), 1.9–1.96 (m, 2H), 2.65 (t, 2H, -CH_2_-N^1^-, *J* = 5.4 Hz), 2.84–2.86 (m, 1H), 2.89 (bs, 4H, 2CH_2_ pip., *J* = 4.5 Hz), 3.18 (s, 2H), 3.23 (bs, 4H, 2CH_2_ pip., *J* = 4.7 Hz), 3.31–3.35 (m, 2H, -NH-CH_2_-), 3.86 (s, 3H, -OCH_3_), 5.97 (dd, 1H, *J* = 5.5, 2.5 Hz), 6.20 (dd, 1H, *J* = 5.5, 2.6 Hz), 6.82 (bs, 1H, NH), 6.86–6.88 (m, 2H), 6.90–6.94 (m, 1H), 7.01–7.05 (m, 1H); ^13^C-NMR (101 MHz, CDCl_3_) δ: 24.70, 29.72, 38.40, 42.66, 44.85, 46.14, 49.65, 49.89, 53.26, 55.39, 56.75, 111.24, 118.34, 121.06, 123.48, 132.39, 137.56, 140.36, 152.17, 174.46; ESI-MS *m/z* [M+H]^+^ calculated for C_22_H_31_N_3_O_2_ 369.50, Found = 370.3; Anal. Calcd for C_22_H_31_N_3_O_2_: C, 71.51; H, 8.46; N, 11.37. Found C, 71.36; H, 8.49; N, 11.38.

#### 4.1.7. Synthesis of Exo-N-(3-(4-(4-methoxyphenyl) piperazin-1-yl)propyl)bicyclo[2.2.1]hept-5-ene-2-carboxamide (**Norbo-7**) and endo-N-(3-(4-(4-methoxyphenyl) piperazin-1-yl) propyl) bicyclo[2.2.1]hept-5-ene-2-carboxamide (**Norbo-8**)

Following the synthetic procedure reported above, **Norbo-7** and **Norbo-8** were synthetized starting from **2** (1.00 g; 4.70 mmol) and 1-(4-methoxyphenyl) piperazine (0.90 g; 4.70 mmol).

**Norbo-7:** Yield: 33%; mp: 119.3–120.1 °C; ^1^H-NMR (400 MHz, CDCl_3_) δ: 1.27 (d, 1H, *J* = 2.0 Hz), 1.32 (d, 1H, *J* = 8.3 Hz), 1.68 (d, 1H, *J* = 8.3 Hz), 1.77–1.80 (m, 2H), 1.90–1.93 (m, 1H), 1.97–1.99 (m, 1H), 2.60 (t, 2H, -CH_2_-N^1^- *J* = 6.1 Hz), 2.74 (bs, 4H, 2CH_2_ pip., *J* = 4.6 Hz), 2.89 (s, 1H), 2.93 (s, 1H), 3.16 (bs, 4H, 2CH_2_ pip., *J* = 4.7 Hz), 3.38–3.39 (m, 2H, -NH-CH_2_-), 3.77 (s, 3H, -OCH_3_), 6.07 (dd, 1H, *J* = 5.3, 2.7 Hz), 6.11 (dd, 1H, *J* = 5.5, 2.8 Hz), 6.83 (d, 2H, *J* = 8.9 Hz), 6.89 (d, 2H, *J* = 8.9 Hz), 7.05 (bs, 1H, NH). ^13^C-NMR (101 MHz, CDCl_3_) δ: 24.92, 30.36, 39.22, 41.55, 44.87, 46.35, 47.24, 50.39, 53.25, 55.54, 57.28, 114.49, 118.39, 135.96, 138.19, 145.16, 154.12, 175.54; ESI-MS *m/z* [M+H]^+^ calculated for C_22_H_31_N_3_O_2_ 369.50, Found = 370.4; Anal. Calcd for C_22_H_31_N_3_O_2_: C, 71.51; H, 8.46; N, 11.37. Found C, 71.65; H, 8.49; N, 11.41.

**Norbo-8:** Yield: 36%; mp: 115.1–116.2 °C.^1^H-NMR (400 MHz, CDCl_3_) δ: 1.26 (d, 1H, *J* = 3.3 Hz), 1.34 (dd, 1H, *J* = 5.4, 3.0 Hz), 1.41 (d, 1H, *J* = 6.9 Hz), 1.74–1.77 (m, 1H), 1.89–1.93 (m, 2H), 2.58 (t, 2H, -CH_2_-N^1^-, *J* = 5.4 Hz), 2.73 (bs, 4H, 2CH_2_ pip., *J* = 4.7 Hz), 2.84–2.86 (m, 1H), 2.89 (s, 2H), 3.18 (bs, 4H, 2CH_2_ pip., *J* = 4.7 Hz), 3.32–3.33 (m, 2H, -NH-CH_2_-), 3.77 (s, 3H, -OCH_3_), 5.98 (dd, 1H, *J* = 5.5, 2.7 Hz), 6.20 (dd, 1H, *J* = 5.5, 2.6 Hz), 6.66 (bs, 1H, NH), 6.84 (d, 2H, *J* = 8.7 Hz), 6.90 (d, 2H, *J* = 8.7 Hz). ^13^C-NMR (101 MHz, CDCl_3_) δ: 25.16, 29.80, 38.94, 42.65, 44.89, 46.12, 49.87, 50.38, 53.37, 55.55, 57.21, 114.48, 118.33, 132.41, 137.52, 145.26, 154.05, 174.22;

ESI-MS *m/z* [M+H] ^+^ calculated for C_22_H_31_N_3_O_2_ 369.50, Found = 370.3; Anal. Calcd for C_22_H_31_N_3_O_2_: C, 71.51; H, 8.46; N, 11.37. Found C, 71.66; H, 8.48; N, 11.38.

#### 4.1.8. Synthesis of Exo-N-(3-(4-(2-chlorophenyl) piperazin-1-yl) propyl) bicyclo[2.2.1]hept-5-ene-2-carboxamide (**Norbo-9**) and endo-N-(3-(4-(2-chlorophenyl) piperazin-1-yl)propyl)bicyclo[2.2.1]hept-5-ene-2-carboxamide (**Norbo-10**)

Following the synthetic procedure reported above, **Norbo-9** and **Norbo-10** were synthetized starting from **2** (1.00 g; 4.70 mmol) and 1-(2-chlorophenyl) piperazine hydrochloride (1.10 g; 4.70 mmol), in the presence of two equivalents of K_2_CO_3_ (1.30 g; 9.4 mmol). The final isomer **Norbo-10** was converted into the corresponding hydrochloride salt.

**Norbo-9:** Yield: 34%; mp: 77.6–80.0 °C; ^1^H-NMR (400 MHz, CDCl_3_) δ: 1.32–1.34 (m, 2H), 1.69 (d, 1H, *J* = 8.1 Hz), 1.81 (m, 2H), 1.92–1.95 (m, 1H), 2.02 (m, 1H), 2.66 (t, 2H, -CH_2_-N^1^-, *J* = 6.1 Hz), 2.80 (bs, 4H, 2CH_2_ pip., *J* = 4.5 Hz), 2.91 (s, 1H), 2.95 (s, 1H), 3.17 (bs, 4H, 2CH_2_ pip., *J* = 4.5 Hz), 3.40 (m, 2H, -NH-CH_2_-), 6.10 (dd, 1H, *J* = 5.5, 2.4 Hz), 6.13 (dd,1H, *J* = 5.5, 2.6 Hz), 7.00–7.05 (m, 3H), 7.24 (t, 1H, *J* = 7.3 Hz), 7.36 (d, 1H, *J* = 7.6 Hz). ^13^C-NMR (101 MHz, CDCl_3_) δ: 24.87, 30.37. 39.03. 41.56, 44.89, 46.36, 47.24, 50.70, 53.27, 57.13, 119.98, 123.72, 127.68, 130.70, 134.18, 135.97, 138.02, 168.88, 175.57; ESI-MS *m/z* [M+H]^+^ calculated for C_21_H_28_ClN_3_O 373.92, Found = 374.2; Anal. Calcd for C_21_H_28_ClN_3_O: C, 67.45; H, 7.55; N, 11.24. Found C, 67.46; H, 7.57; N, 11.27.

**Norbo-10:** Yield: 37%; mp: 93.4–96.1 °C; ^1^H-NMR (400 MHz, CDCl_3_) δ: 1.24–1.26 (d, 1H, *J* = 3.0 Hz), 1.29–1.32 (m, 2H), 1.70–1.75 (m, 1H), 1.83 (m, 2H), 2.48 (t, 2H, -CH_2_-N^1^-, *J* = 5.1 Hz), 2.80 (bs, 4H, 2CH_2_ pip., *J* = 4.7 Hz), 3.08 (m, 1H), 3.15 (s, 1H), 3.31 (bs, 4H, 2CH_2_ pip., *J* = 4.7 Hz), 3.41 (m, 1H), 3.53 (m, 2H, -NH-CH_2_-), 5.86 (dd, 1H, *J* = 5.4, 2.6 Hz), 6.10 (dd, 1H, *J* = 5.5, 3.0 Hz), 7.08 (t, 1H, *J* = 7.0 Hz), 7.18 (d, 1H, *J* = 7.3 Hz), 7.31 (t, 1H, *J* = 7.3 Hz), 7.43 (d, 1H, *J* = 7.3 Hz), 7.81 (bs, 1H, NH); ^13^C-NMR (101 MHz, CDCl_3_) δ: 24.26, 28.82, 36.28, 42.53, 43.84, 46.01, 48.12, 49.84, 51.65, 53.96, 121.46, 125.28, 127.98, 128.72, 130.92, 132.75, 137.38, 147.84, 173.63; ESI-MS *m/z* [M+H]^+^ calculated for C_21_H_28_ClN_3_O 373.92, Found = 374.0; Anal. Calcd for C_21_H_28_ClN_3_O: C, 67.45; H, 7.55; N, 11.24. Found C, 67.47; H, 7.56; N, 11.27.

#### 4.1.9. Synthesis of Exo-N-(3-(4-(4-chlorophenyl) piperazin-1-yl)propyl)bicyclo[2.2.1]hept-5-ene-2-carboxamide (**Norbo-11**) and endo-N-(3-(4-(4-chlorophenyl) piperazin-1-yl)propyl)bicyclo[2.2.1]hept-5-ene-2-carboxamide (**Norbo-12**)

Following the synthetic procedure reported above, **Norbo-11** and **Norbo-12** were synthetized starting from **2** (1.00 g; 4.70 mmol) and 1-(4-chlorophenyl) piperazine (0.92 g; 4.70 mmol).

**Norbo-11:** Yield: 34%; mp: 144.7–146.2 °C; ^1^H-NMR (400 MHz, CDCl_3_) δ: 1.27–1.29 (m, 1H), 1.33 (d, 1H, *J* = 8.3 Hz), 1.68 (d, 1H, *J* = 8.3 Hz), 1.76–1.79 (m, 2H), 1.90–1.92 (m, 1H), 1.96–1.98 (m, 1H), 2.56 (t, 2H, -CH_2_-N^1^-, *J* = 6.3 Hz), 2.68 (bs, 4H, 2CH_2_ pip., *J* = 4.7 Hz), 2.90 (s, 1H), 2.92 (s, 1H), 3.19 (bs, 4H, 2CH_2_ pip., *J* = 4.7 Hz), 3.34–3.36 (m, 2H, -NH-CH_2_-), 6.05 (dd, 1H, *J* = 5.4, 3.0 Hz), 6.11 (dd, 1H, *J* = 5.5, 2.9 Hz), 6.83 (d, 2H, *J* = 8.9 Hz), 6.92 (bs, 1H, NH), 7.20 (d, 2H, *J* = 8.9 Hz); ^13^C-NMR (101 MHz, CDCl_3_) δ: 25.11, 30.39, 39.28, 41.56, 44.93, 46.38, 47.25, 49.11, 53.07, 57.32, 117.36, 124.97, 129.06, 135.95, 138.28, 149.59, 175.52; ESI-MS *m/z* [M+H]^+^ calculated for C_21_H_28_ClN_3_O 373.92, Found = 374.2; Anal. Calcd for C_21_H_28_ClN_3_O: C, 67.45; H, 7.55; N, 11.24. Found C, 67.71; H, 7.58; N, 11.28.

**Norbo-12:** Yield: 38%; mp: 147.8–148.9 °C; ^1^H-NMR (400 MHz, CDCl_3_) δ: 1.26 (d, 1H, *J* = 8.2 Hz), 1.32 (dd, 1H, *J* = 8.5, 3.0 Hz), 1.41 (d, 1H, *J* = 6.9 Hz), 1.70–1.77 (m, 1H), 1.87–1.93 (m, 2H), 2.51 (t, 2H, -CH_2_-N^1^-, *J* = 5.4 Hz), 2.66 (bs, 4H, 2CH_2_ pip., *J* = 4.7 Hz), 2.83–2.86 (m, 1H), 2.89 (s, 1H), 3.14 (s, 1H), 3.21 (bs, 4H, 2CH_2_ pip., *J* = 4.7 Hz) 3.30–3.33 (m, 2H, -NH-CH_2_-), 5.97 (dd, 1H, *J* = 5.2, 2.5 Hz), 6.20 (dd,1H, *J* = 5.5, 3.1 Hz), 6.48 (bs, 1H, NH), 6.83 (d, 2H, *J* = 8.8 Hz), 7.20 (d, 2H, *J* = 8.8 Hz); ^13^C-NMR (101 MHz, CDCl_3_) δ: 25.42, 29.84, 38.93, 42.68, 44.93, 46.14, 49.06, 49.91, 53.16, 57.17, 117.32, 124.87, 129.04, 132.43, 137.60, 149.64, 174.24; ESI-MS *m/z* [M+H]^+^ calculated for C_21_H_28_ClN_3_O 373.92, Found = 374.1; Anal. Calcd for C_21_H_28_ClN_3_O: C, 67.45; H, 7.55; N, 11.24. Found C, 67.46; H, 7.56; N, 11.23.

#### 4.1.10. Synthesis of Exo-N-(3-(4-(3,4-dichlorophenyl)piperazin-1-yl)propyl)bicyclo[2.2.1]hept-5-ene-2-carboxamide (**Norbo-13**) and endo-N-(3-(4-(3,4-dichlorophenyl)piperazin-1-yl)propyl)bicyclo[2.2.1]hept-5-ene-2-carboxamide (**Norbo-14**)

Following the synthetic procedure reported above, **Norbo-13** and **Norbo-14** were synthetized starting from **2** (1.00 g; 4.70 mmol) and 1-(3,4-dichlorophenyl) piperazine (1.09 g; 4.70 mmol).

**Norbo-13:** Yield: 32%; mp: 123.6–124.7 °C; ^1^H-NMR (400 MHz, CDCl_3_) δ: 1.28–1.35 (m, 2H), 1.67 (d, 1H, *J* = 8.1 Hz), 1.79–1.82 (m, 2H), 1.89–1.92 (m, 1H), 1.96–1.99 (m, 1H), 2.62 (t, 2H, -CH_2_-N^1^-, *J* = 6.5 Hz), 2.74 (bs, 4H, 2CH_2_ pip., *J* = 4.7 Hz), 2.91(s, 2H), 3.26 (bs, 4H, 2CH_2_ pip., *J* = 4.7 Hz) 3.36–3.40 (m, 2H, -NH-CH_2_-), 6.06 (dd, 1H, *J* = 5.5, 2.6 Hz), 6.11(dd, 1H, *J* = 5.4, 2.8 Hz), 6.72 (dd, 1H, *J* = 8.9, 2.1 Hz), 6.83 (bs, 1H, NH), 6.95 (d, 1H, *J* = 8.7 Hz), 7.27 (d, 1H, *J* = 8.0 Hz); ^13^C-NMR (101 MHz, CDCl_3_) δ: 25.02, 30.42, 38.75, 41.54, 44.84, 46.36, 47.23, 48.29, 52.71, 56.87, 115.54, 117.57, 130.56, 132.89, 135.47, 138.26, 152.11, 175.71; ESI-MS *m/z* [M+H]^+^ calculated for C_21_H_27_Cl_2_N_3_O 408.36, Found = 408.0; Anal. Calcd for C_21_H_27_Cl_2_N_3_O: C, 61.76; H, 6.66; N, 10.29. Found C, 61.51; H, 6.64; N, 10.24.

**Norbo-14:** Yield: 42%; mp: 99.1–101.3 °C; ^1^H-NMR (400 MHz, CDCl_3_) δ: 1.27 (d, 1H, *J* = 8.1 Hz), 1.32 (dd, 1H, *J* = 8.3, 3.0 Hz), 1.43 (d, 1H, *J* = 6.5 Hz), 1.73–1.77 (m, 1H), 1.89–1.91 (m, 2H), 2.56 (t, 2H, -CH_2_-N^1^-, *J* = 5.5 Hz), 2.71 (bs, 4H, 2CH_2_ pip., *J* = 4.7 Hz), 2.84–2.86 (m, 1H), 2.90 (s, 1H), 3.14 (s, 1H), 3.26 (bs, 4H, 2CH_2_ pip., *J* = 4.7 Hz) 3.30–3.32 (m, 2H, -NH-CH_2_-), 5.98 (dd, 1H, *J* = 5.4, 2.7 Hz), 6.22 (dd, 1H, *J* = 5.4, 3.0 Hz), 6.43 (bs, 1H, NH), 6.73 (d, 1H, *J* = 8.8 Hz), 6.96 (s, 1H), 7.27 (d, 1H, *J* = 8.7 Hz); ^13^C-NMR (101 MHz, CDCl_3_) δ: 25.32, 29.81, 38.49, 42.66, 44.89, 46.12, 48.31, 49.91, 52.81, 56.74, 115.46, 117.49, 130.52, 132.35, 135.37, 137.62, 154.37, 174.31; ESI-MS *m/z* [M+H]^+^ calculated for C_21_H_27_Cl_2_N_3_O 408.36, Found = 408.4; Anal. Calcd for C_21_H_27_Cl_2_N_3_O: C, 61.76; H, 6.66; N, 10.29. Found C, 62.00; H, 6.68; N, 10.33.

#### 4.1.11. Synthesis of Exo-N-(3-(4-(2-fluorophenyl)piperazin-1-yl)propyl)bicyclo[2.2.1]hept-5-ene-2-carboxamide (**Norbo-15**) and endo.-N-(3-(4-(2-fluorophenyl)piperazin-1-yl)propyl)bicyclo[2.2.1]hept-5-ene-2-carboxamide (**Norbo-16**)

Following the synthetic procedure reported above, **Norbo-15** and **Norbo-16** were synthetized starting from **2** (1.00 g; 4.70 mmol) and 1-(2-fluorophenyl) piperazine (0.85 g; 4.70 mmol).

**Norbo-15**: Yield: 33% mp: 82.8–84.1 °C; ^1^H-NMR (400 MHz, CDCl_3_) δ: 1.29–1.43 (m, 2H), 1.68 (d, 1H, *J* = 8.2 Hz), 1.79 -1.85 (m, 2H), 1.91–1.94 (m, 1H), 2.00–2.02 (m, 1H), 2.68 (t, 2H, -CH_2_-N^1^-, *J* = 6.4 Hz), 2.83 (bs, 4H, 2CH_2_ pip., *J* = 4.9 Hz), 2.90 (s, 1H), 2.94 (s, 1H), 3.23 (bs, 4H, 2CH_2_ pip., *J* = 4.8 Hz), 3.39–3.41 (m, 2H, -NH-CH_2_-), 6.09 (dd, 1H, *J* = 5.4, 2.6 Hz), 6.12 (dd, 1H, *J* = 5.7, 2.9 Hz), 6.94–6.99 (m, 2H), 7.01–7.07 (m, 3H); ^13^C-NMR (101 MHz, CDCl_3_) δ: 24.77, 30.39, 38.32, 41.56, 44.85, 46.35, 47.24, 49.79, 53.11, 56.97, 116.10, 119.00, 123.06, 124.53, 135.97, 138.18, 154.46, 156.90, 177.52; ESI-MS *m/z* [M+H]^+^ calculated for C_21_H_28_FN_3_O 357.46, Found = 358.2; Anal. Calcd for C_21_H_28_FN_3_O: C, 70.56; H, 7.90; N, 11.76. Found C, 70.77; H, 7.87; N, 11.72.

**Norbo-16:** Yield: 44%; mp: 88.5–89.1 °C; ^1^H-NMR (400 MHz, CDCl_3_) δ: 1.28 (d, 1H, *J* = 8.2 Hz), 1.34–1.37 (m, 1H), 1.43 (d, 1H, *J* = 6.2 Hz), 1.74–1.77 (m, 1H), 1.89–1.95 (m, 2H), 2.57 (t, 2H, -CH_2_-N^1^-, *J* = 5.4 Hz), 2.75 (bs, 4H, 2CH_2_ pip., *J* = 4.7 Hz), 2.85–2.88 (m, 1H), 2.90 (s, 1H), 3.17 (s, 1H), 3.20 (bs, 4H, 2CH_2_ pip., *J* = 4.7 Hz), 3.30–3.35 (m, 2H, -NH-CH_2_-), 5.98 (dd, 1H, *J* = 5.5, 2.7 Hz), 6.20(dd, 1H, *J* = 5.7, 3.2 Hz), 6.62 (bs, 1H, NH), 6.93–6.97 (m, 2H), 7.01–7.09 (m, 2H); ^13^C-NMR (101 MHz, CDCl_3_) δ: 25.12, 29.80, 38.76, 42.67, 44.90, 46.13, 49.89, 50.08, 53.27, 57.04, 116.07, 118.95, 122.86, 124.49, 132.40, 137.55, 154.49, 156.93, 174.25; ESI-MS *m/z* [M+H]^+^ calculated for C_21_H_28_FN_3_O 357.46, Found = 358.2; Anal. Calcd for C_21_H_28_FN_3_O: C, 70.56; H, 7.90; N, 11.76. Found C, 70.27; H, 7.88; N, 11.71.

#### 4.1.12. Synthesis of Exo-N-(3-(4-(4-fluorophenyl)piperazin-1-yl)propyl)bicyclo[2.2.1]hept-5-ene-2-carboxamide (**Norbo-17**) and endo.-N-(3-(4-(4-fluorophenyl)piperazin-1-yl)propyl)bicyclo[2.2.1]hept-5-ene-2-carboxamide (**Norbo-18**)

Following the synthetic procedure reported above, **Norbo-17** and **Norbo-18** were synthetized starting from **2** (1.00 g; 4.70 mmol) and 1-(4-fluorophenyl) piperazine (0.85 g; 4.70 mmol).

**Norbo-17:** Yield: 32%; mp: 105.5–106.2 °C. ^1^H-NMR (400 MHz, CDCl_3_) δ: 1.28 (d, 1H, *J* = 8.1 Hz), 1.33 (d, 1H, *J* = 7.9 Hz), 1.68 (d, 1H, *J* = 8.3 Hz), 1.79–1.81 (m, 2H), 1.90–1.93 (m, 1H), 1.98–2.00 (m, 1H), 2.63 (t, 2H, -CH_2_-N^1^-, *J* = 6.0 Hz), 2.75 (bs, 4H, 2CH_2_ pip., *J* = 4.7 Hz), 2.90 (s, 1H), 2.93 (s, 1H), 3.20 (bs, 4H, 2CH_2_ pip., *J* = 4.7 Hz), 3.37–3.41 (m, 2H, -NH-CH_2_-), 6.07 (dd, 1H, *J* = 5.2, 2.6 Hz), 6.12 (dd, 1H, *J* = 5.4, 2.8 Hz), 6.88 (d, 2H, *J* = 8.2 Hz), 6.95 (d, 2H, *J* = 8.2 Hz), 6.99 (bs, 1H, NH). ^13^C-NMR (101 MHz, CDCl_3_) δ: 25.00, 30.38, 39.00, 41.55, 44.87, 46.35, 47.23, 49.87, 53.10, 57.07, 115.53, 118.07, 135.93, 138.21, 148.54, 175.57; ESI-MS *m/z* [M+H]^+^ calculated for C_21_H_28_FN_3_O 357.46, Found = 358.2; Anal. Calcd for C_21_H_28_FN_3_O: C, 70.56; H, 7.90; N, 11.76. Found C, 70.41; H, 7.91; N, 11.74.

**Norbo-18:** Yield: 37%; mp: 112.7–113.5 °C; ^1^H-NMR (400 MHz, CDCl_3_) δ: 1.26 (d, 1H, *J* = 8.1 Hz), 1.33 (dd, 1H, *J* = 8.2, 3.1 Hz), 1.42 (d, 1H, *J* = 6.6 Hz), 1.73–1.76 (m, 1H), 1.88–1.94 (m, 2H), 2.53 (t, 2H, -CH_2_-N^1^-, *J* = 6.0 Hz), 2.70 (bs, 4H, 2CH_2_ pip., *J* = 4.5 Hz), 2.83–2.86 (m, 1H), 2.90 (s, 1H), 3.15 (s, 1H), 3.19 (bs, 4H, 2CH_2_ pip., *J* = 4.5 Hz), 3.30–3.34 (m, 2H, -NH-CH_2_-), 5.98 (dd, 1H, *J* = 5.4, 2.9 Hz), 6.21(dd, 1H, *J* = 5.4, 2.5 Hz), 6.54 (bs, 1H, NH), 6.88 (d, 2H, *J* = 8.2 Hz), 6.95 (d, 2H, *J* = 8.2 Hz); ^13^C-NMR (101 MHz, CDCl_3_) δ: 25.28, 29.80, 38.80, 42.65, 44.89, 46.12, 49.88, 49.91, 53.23, 57.04, 115.49, 117.95, 132.39, 137.56, 147.58, 156.16, 174.23; ESI-MS *m/z* [M+H]^+^ calculated for C_21_H_28_FN_3_O 357.46, Found = 358.2; Anal. Calcd for C_21_H_28_FN_3_O: C, 70.56; H, 7.90; N, 11.76. Found C, 70.53; H, 7.86; N, 11.75.

#### 4.1.13. Synthesis of Exo-N-(3-(4-(3-(trifluoromethyl)phenyl)piperazin-1-yl)propyl)bicyclo[2.2.1]hept-5-ene-2-carboxamide (**Norbo-19**) and endo-N-(3-(4-(3-(trifluoromethyl)phenyl)piperazin-1-yl)propyl)bicyclo[2.2.1]hept-5-ene-2-carboxamide (**Norbo-20**)

Following the synthetic procedure reported above, **Norbo-19** and **Norbo-20** were synthetized starting from **2** (1.00 g; 4.70 mmol) and 1-(3-(trifluoromethyl) phenyl) piperazine (1.08 g; 4.70 mmol). The final isomer, **Norbo-19,** was converted into the corresponding hydrochloride salt.

**Norbo-19:** Yield: 35%; mp: 164.7–166.0 °C; ^1^H-NMR (400 MHz, DMSO-*d*_6_) δ: 1.13 (m, 2H), 1.61 (d, 1H, *J* = 8.1 Hz), 1.74–1.79 (m, 2H), 1.83–1.86 (m, 1H), 2.00–2.04 (m, 1H), 2.81 (bs, 4H, 2CH_2_ pip., *J* = 4.7 Hz), 3.09–3.11 (m, 8H), 3.51–3.54 (m, 2H, -NH-CH_2_-), 6.11 (dd, 1H, *J* = 5.5, 2.7 Hz), 6.12 (dd, 1H, *J* = 5.9, 2.9 Hz), 7.13 (d, 1H, *J* = 7.3 Hz), 7.25–7.28 (m, 2H), 7.43 (t, 1H, *J* = 8.0 Hz), 8.04 (bs, 1H, NH); ^13^C-NMR (101 MHz, DMSO-*d*_6_) δ: 24.22, 30.29, 36.40, 41.42, 43.50, 45.28, 46.13, 47.28, 50.88, 53.89, 112.10, 116.21, 119.70, 130.13, 130.64, 136.66, 138.24, 150.29, 175.35; ESI-MS *m/z* [M+H]^+^ calculated for C_22_H_28_F_3_N_3_O 407.47, Found = 408.2; Anal. Calcd for C_22_H_28_F_3_N_3_O: C, 64.85; H, 6.93; N, 10.31. Found C, 64.59; H, 6.95; N, 10.35.

**Norbo-20:** Yield: 38%; mp. 79.4–81.7 °C; ^1^H-NMR (400 MHz, CDCl_3_) δ: 1.27 (d, 1H, *J* = 8.1 Hz), 1.32 (dd, 1H, *J* = 8.2, 2.7 Hz), 1.42 (d, 1H, *J* = 6.5 Hz), 1.74–1.77 (m, 1H), 1.89–1.94 (m, 2H), 2.56 (t, 2H, -CH_2_-N^1^-, *J* = 5.5 Hz), 2.72 (bs, 4H, 2CH_2_ pip., *J* = 4.7 Hz), 2.83–2.86 (m, 1H), 2.90 (s, 1H), 3.14 (s, 1H), 3.32 (bs, 6H), 5.98 (dd, 1H, *J* = 5.4, 2.9 Hz), 6.21(dd, 1H, *J* = 5.5, 3.1 Hz), 6.46 (bs, 1H, NH), 7.05–7.11 (m, 2H), 7.25 (d, 1H, *J* = 7.3 Hz), 7.34–7.37 (m, 1H); ^13^C-NMR (101 MHz, CDCl_3_) δ: 25.33, 29.81, 38.60, 42.66, 44.89, 46.13, 48.36, 49.90, 52.96, 56.86, 112.38, 116.12, 118.83, 129.31, 129.63, 132.36, 137.60, 150.94, 174.28; ESI-MS *m/z* [M+H]^+^ calculated for C_22_H_28_F_3_N_3_O 407.47, Found = 408.3; Anal. Calcd for C_22_H_28_F_3_N_3_O: C, 64.85; H, 6.93; N, 10.31. Found C, 65.10; H, 6.53; N, 10.28.

#### 4.1.14. Synthesis of Exo-N-(3-(4-(pyridin-2-yl)piperazin-1-yl)propyl)bicyclo[2.2.1]hept-5-ene-2-carboxamide (**Norbo-21**) endo-N-(3-(4-(pyridin-2-yl)piperazin-1-yl)propyl)bicyclo[2.2.1]hept-5-ene-2-carboxamide (**Norbo-22**)

Following the synthetic procedure reported above, **Norbo-21** and **Norbo-22** were synthetized starting from **2** (1.00 g; 4.70 mmol) and 1-(pyridin-2-yl) piperazine (0.77 g; 4.70 mmol).

Yield: 44%; mp: 97.1–98.3 °C; ^1^H-NMR (400 MHz, CDCl_3_) δ: 1.27–1.35 (m, 2H), 1.69–1.75 (m, 3H), 1.90–1.96 (m, 2H), 2.51 (t, 2H, -CH_2_-N^1^-, *J* = 6.1 Hz), 2.59 (bs, 4H, 2CH_2_ pip., *J* = 4.9 Hz), 2.89 (s, 1H), 2.93 (s, 1H), 3.36–3.39 (m, 2H, -NH-CH_2_-), 3.55 (bs, 4H, 2CH_2_ pip., *J* = 4.9 Hz) 6.06 (dd, 1H, *J* = 5.3, 2.9 Hz), 6.10 (dd, 1H, *J* = 5.5, 2.7 Hz), 6.64–6.66 (m, 2H), 6.95 (bs, 1H, NH), 7.47 (t, 1H, *J* = 7.1 Hz), 8.20 (d, 1H, J = 5.5 Hz); ^13^C-NMR (101 MHz, CDCl_3_) δ: 25.00, 29.76, 38.62, 42.66, 44.68, 44.88, 46.12, 49.89, 52.90, 57.03, 107.16, 113.80, 132.40, 137.57, 137.62, 148.01, 159.04, 174.33; ESI-MS *m/z* [M+H]^+^ calculated for C_20_H_28_N_4_O 340.46, Found = 341.4; Anal. Calcd for C_20_H_28_N_4_O: C, 70.56; H, 8.29; N, 16.46. Found C, 70.27; H, 8.26; N, 16.39.

**Norbo-22:** Yield: 45%; mp: 105.3–107.0 °C; ^1^H-NMR (400 MHz, CDCl_3_) δ: 1.26 (d, 1H, *J* = 8.1 Hz), 1.33 (d, 1H, *J* = 8.3), 1.41 (d, 1H, *J* = 6.8 Hz), 1.72–1.76 (m, 1H), 1.88–1.93 (m, 2H), 2.52 (t, 2H, -CH_2_-N^1^-, J = 5.7 Hz), 2.59 (bs, 4H, 2CH_2_ pip., *J* = 4.9 Hz), 2.83–2.85 (m, 1H), 2.89 (s, 1H), 3.14 (s, 1H), 3.31–3.33 (m, 2H, -NH-CH_2_-), 3.62 (bs, 4H, 2CH_2_ pip., *J* = 4.9 Hz), 5.98 (dd, 1H, *J* = 5.5, 2.9 Hz), 6.20 (dd, 1H, *J* = 5.4, 3.2 Hz), 6.64–6.66 (m, 3H), 7.47 (t, 1H, *J* = 7.3 Hz), 8.20 (d, 1H, *J* = 5.8 Hz); ^13^C-NMR (101 MHz, CDCl_3_) δ: 25.18, 29.79, 38.93, 42.65, 44.65, 44.92, 46.11, 49.88, 53.05, 57.30, 107.10, 113.63, 132.41, 137.53, 147.97, 159.18, 174.21; ESI-MS *m/z* [M+H]^+^ calculated for C_20_H_28_N_4_O 340.46, Found = 341.2; Anal. Calcd for C_20_H_28_N_4_O: C, 70.56; H, 8.29; N, 16.46. Found C, 70.55; H, 8.27; N, 16.44.

#### 4.1.15. Synthesis of Exo-N-(3-(4-(pyrimidin-2-yl)piperazin-1-yl)propyl)bicyclo[2.2.1]hept-5-ene-2-carboxamide (**Norbo-23**) and endo-N-(3-(4-(pyrimidin-2-yl)piperazin-1-yl)propyl)bicyclo[2.2.1]hept-5-ene-2-carboxamide (**Norbo-24**)

Following the synthetic procedure reported above, **Norbo-23** and **Norbo-24** were synthetized starting from **2** (1.00 g; 4.70 mmol) and 2-(piperazin-1-yl) pyrimidine (0.77 g; 4.70 mmol).

**Norbo-23:** Yield: 43%; mp: 107.6–108.8 °C; ^1^H-NMR (400 MHz, CDCl_3_) δ: 1.29–1.35 (m, 2H), 1.69 (d, 1H, *J* = 8.0 Hz), 1.79–1.82 (m, 2H), 1.89–1.92 (m, 1H), 1.98–2.00 (m, 1H), 2.60–2.64 (m, 6H), 2.90 (s, 1H), 2.93 (s, 1H), 3.37–3.41 (m, 2H, -NH-CH_2_-), 3.92 (bs, 4H, 2CH_2_ pip., *J* = 4.7 Hz), 6.09 (dd, 1H, *J* = 5.5, 2.7 Hz), 6.11 (dd, 1H, *J* = 5.5, 2.9 Hz), 6.51 (t, 1H, *J* = 8.0 Hz), 6.98 (bs, 1H, NH), 8.30 (d, 2H, *J* = 7.6 Hz); ^13^C-NMR (101 MHz, CDCl_3_) δ: 24.93, 30.48, 39.01, 41.54, 43.15, 44.86, 46.38, 47.16, 52.96, 57.25, 110.33, 136.05, 138.14, 157.77, 161.44, 175.65; ESI-MS *m/z* [M+H]^+^ calculated for C_19_H_27_N_5_O 341.45, Found = 342.2; Anal. Calcd for C_19_H_27_N_5_O: C, 66.83; H, 7.97; N, 20.51. Found C, 66.62; H, 7.99; N, 20.44.

**Norbo-24:** Yield: 35%; mp: 116.5–118.9 °C; ^1^H-NMR (400 MHz, CDCl_3_) δ: 1.27 (d, 1H, *J* = 8.2 Hz), 1.34 (dd, 1H, *J* = 8.3, 2.7 Hz), 1.42 (d, 1H, *J* = 6.5 Hz), 1.74–1.79 (m, 1H), 1.89–1.95 (m, 2H), 2.54 (t, 2H, -CH_2_-N^1^-, *J* = 5.7 Hz), 2.63 (bs, 4H, 2CH_2_ pip., *J* = 5.0 Hz), 2.84–2.87 (m, 1H), 2.90 (s, 1H), 3.16 (s, 1H), 3.31–3.35 (m, 2H, -NH-CH_2_-), 3.93 (bs, 4H, 2CH_2_ pip., *J* = 5.0 Hz), 5.99 (dd, 1H, *J* = 5.6, 3.1 Hz), 6.20 (dd, 1H, *J* = 5.7, 3.2 Hz), 6.51 (t, 1H, *J* = 8.0 Hz), 6.66 (bs, 1H, NH), 8.31 (d, 2H, *J* = 7.6 Hz); ^13^C-NMR (101 MHz, CDCl_3_) δ: 25.06, 29.77, 38.72, 42.66, 43.13, 44.90, 46.13, 49.90, 53.04, 57.16, 110.28, 132.41, 137.56, 157.75, 161.44, 172.98; ESI-MS *m/z* [M+H]^+^ calculated for C_19_H_27_N_5_O 341.45, Found = 342.2; Anal. Calcd for C_19_H_27_N_5_O: C, 66.83; H, 7.97; N, 20.51. Found C, 66.76; H, 7.96; N, 20.46.

#### 4.1.16. Synthesis of Exo-N-(3-(4-(benzo[d][1,3]dioxol-5-ylmethyl)piperazin-1-yl)propyl)bicyclo[2.2.1]hept-5-ene-2-carboxamide (**Norbo-25**) and endo-N-(3-(4-(benzo[d][1,3]dioxol-5-ylmethyl)piperazin-1-yl)propyl)bicyclo[2.2.1]hept-5-ene-2-carboxamide (**Norbo-26**)

Following the synthetic procedure reported above, **Norbo-25** and **Norbo-26** were synthetized starting from **2** (1.00 g; 4.70 mmol) and 1-(benzo[d][1,3] dioxol-5-ylmethyl)piperazine (1.03 g; 4.70 mmol). The final isomer **Norbo-25** was converted into the corresponding hydrochloride salt.

**Norbo-25:** Yield: 33%; mp: 215.5–217.5 °C; ^1^H-NMR (400 MHz, CDCl_3_) δ: 1.27–1.35 (m, 2H), 1.66–1.71 (m, 3H), 1.90–1.93 (m, 1H), 1.97–2.01 (m, 1H), 2.50–2.57 (m, 6H), 2.91 (s, 2H), 3.33–3.35 (m, 2H, -NH-CH_2_-), 3.44 (s, 2H, -CH_2_-), 3.48 (bs, 4H, 2CH_2_ pip., *J* = 5.1 Hz), 5.94 (s, 2H, -OCH_2_), 6.10 (dd, 1H, *J* = 5.5, 2.9 Hz), 6.15 (dd, 1H, *J* = 5.5, 3.0 Hz), 6.73 (bs, 2H), 6.83 (s, 1H), 7.18 (bs, 1H, NH); ^13^C-NMR (101 MHz, CDCl_3_) δ: 24.73, 30.31, 38.71, 41.57, 44.89, 46.34, 47.25, 52.41, 53.06, 56.52, 62.54, 100.91, 107.90, 109.42, 122.24, 135.99, 138.71, 146.72, 147.76, 175.44; ESI-MS *m/z* [M+H]^+^ calculated for C_23_H_31_N_3_O_3_ 397.51, Found = 398.1; Anal. Calcd for C_23_H_31_N_3_O_3_: C, 69.49; H, 7.86; N, 10.57. Found C, 69.35; H, 7.83; N, 10.54.

**Norbo-26:** Yield: 34%; mp: 74.9–78.8 °C; ^1^H-NMR (400 MHz, CDCl_3_) δ: 1.27–1.34 (m, 2H), 1.43 (d, 1H, *J* = 6.8 Hz), 1.63–1.66 (m, 1H), 1.86–1.91 (m, 2H), 2.45–2.50 (m, 6H), 2.8–2.84 (m, 1H), 2.90 (s, 1H), 3.14 (s, 1H), 3.26–3.28 (m, 2H, -NH-CH_2_-), 3.43 (s, 2H, -CH_2_-), 3.48 (bs, 4H, 2CH_2_ pip., *J* = 4.7 Hz), 5.93 (s, 2H, -OCH_2_), 5.96 (dd, 1H, *J* = 5.9, 2.8 Hz), 6.19 (dd, 1H, *J* = 5.8, 3.1 Hz), 6.73 (bs, 2H), 6.83 (s, 1H), 7.18 (bs, 1H, NH); ^13^C-NMR (101 MHz, CDCl_3_) δ: 25.09, 29.81, 39.31, 42.65, 44.88, 46.09, 49.81, 52.68, 53.28, 57.47, 62.65, 100.87, 107.84, 109.47, 122.24, 131.59, 132.47, 137.42, 146.63, 147.61, 174.09; ESI-MS *m/z* [M+H]^+^ calculated for C_23_H_31_N_3_O_3_ 397.51, Found = 398.0; Anal. Calcd for C_23_H_31_N_3_O_3_: C, 69.49; H, 7.86; N, 10.57. Found C, 69.35; H, 7.87; N, 10.56.

#### 4.1.17. Synthesis of Exo- N-(3-(4-(furan-2-carbonyl)piperazin-1-yl)propyl)bicyclo[2.2.1]hept-5-ene-2-carboxamide (**Norbo-27**) and endo-N-(3-(4-(furan-2-carbonyl)piperazin-1-yl)propyl)bicyclo[2.2.1]hept-5-ene-2-carboxamide (**Norbo-28**)

Following the synthetic procedure reported above, **Norbo-27** and **Norbo-28** were synthetized starting from **2** (1.00 g; 4.70 mmol) and furan-2-yl(piperazin-1-yl)-methanone (0.85 g; 4.70 mmol). The final isomers **Norbo-27** and **Norbo-28** were converted into the corresponding hydrochloride salts.

**Norbo-27:** Yield: 33%; mp: 211.5–214.2 °C; ^1^H-NMR (400 MHz, DMSO-*d*_6_) δ: 1.13–1.17 (m, 2H), 1.61 (d, 1H, *J* = 8.0 Hz), 1.74–1.76 (m, 2H), 1.82–1.85 (m, 1H), 2.00–2.03 (m, 1H), 2.80 (bs, 4H, 2CH_2_ pip., *J* = 4.7 Hz), 3.05–3.11 (m, 8H), 3.47–3.49 (m, 2H, -NH-CH_2_-), 6.11 (dd, 1H, *J* = 5.7, 2.8 Hz), 6.12 (dd, 1H, *J* = 6.0, 3.0 Hz), 6.65 (t, 1H, *J* = 8.1 Hz), 7.09 (d, 1H, *J* = 8.1 Hz), 7.87 (d, 1H, *J* = 7.3 Hz), 8.04 (bs, 1H, NH); ^13^C-NMR (101 MHz, DMSO-*d*_6_) δ: 24.20, 30.28, 36.32, 40.60, 41.42, 43.49, 46.13, 47.28, 51.11, 54.09, 111.97, 117.03, 136.66, 138.23, 145.74, 146.54, 158.63, 175.31; ESI-MS *m/z* [M+H]^+^ calculated for C_20_H_27_N_3_O_3_ 357.45, Found = 358.2; Anal. Calcd for C_20_H_27_N_3_O_3_: C, 67.20; H, 7.61; N, 11.76. Found C, 67.18; H, 7.59; N, 11.73.

**Norbo-28:** Yield: 38%; mp: 194.3–197.1 °C. ^1^H-NMR (400 MHz, DMSO-*d*_6_) δ: 1.22–1.31 (m, 2H), 1.68–1.75 (m, 3H), 1.76–1.79 (m, 1H), 2.76–2.79 (m, 6H), 3.00–3.08 (m, 6H), 3.13 (s, 1H), 3.46–3.49 (m, 2H, -NH-CH_2_-), 5.82 (dd, 1H, *J* = 5.5, 2.7 Hz), 6.08 (dd, 1H, *J* = 5.6, 3.2 Hz), 6.64 (t, 1H, *J* = 8.1 Hz), 7.08 (d, 1H, *J* = 8.3 Hz), 7.78 (bs, 1H, NH), 7.87 (d, 1H, *J* = 7.3 Hz). ^13^C-NMR (101 MHz, DMSO-*d*_6_) δ: 24.25, 28.73, 36.15, 40.52, 42.50, 43.78, 46.01, 49.84, 51.06, 54.06, 112.00, 117.07, 132.71, 137.42, 145.78, 146.59, 158.60, 173.60; ESI-MS *m/z* [M+H]^+^ calculated for C_20_H_27_N_3_O_3_ 357.45, Found = 358.2; Anal. Calcd for C_20_H_27_N_3_O_3_: C, 67.20; H, 7.61; N, 11.76. Found C, 67.06; H, 7.64; N, 11.79.

### 4.2. In Vitro Receptor Assays

#### 4.2.1. Functional 5-HT_1A_ Receptor Assay

Male Sprague-Dawley rats were decapitated, with their brains removed and placed on ice. Hippocampi were dissected and homogenized with a glass homogenizer in 30 vol. ice-cold TED buffer (50 mM Tris-HCl, 1 mM ethylenediaminetetraacetic acid (EDTA), 1 mM dithiotheritol, pH 7.4). Next, the homogenate was centrifuged at 21,000× *g* for 30 min at 4 °C. The pellet was suspended in 30 vol. TED buffer (pH 7.4) and incubated in a water bath for 10 min at 37 °C to remove endogenous serotonin. The suspension was centrifuged again at 21,000× *g* for 30 min at 4 °C. The pellet was resuspended in 30 vol. TED buffer (pH 7.4) and the centrifugation step was repeated. The final pellet was suspended in 10 vol. 50 mM Tris-HCl (pH 7.4) and stored at −80 °C until use. In the agonist mode, 15 μg/mL of hippocampus homogenate was incubated in triplicate with 0.8 nM [^35^S]GTPγS (guanosine-5′-(γ-thio)-triphosphate) in a assay buffer (50 mM Tris-HCl, pH = 7.4, 1 mM ethylene glycol-bis(β-aminoethyl ether)-N,N,N′,N′-tetraacetic acid tetrasodium salt (EGTA), 3 mM MgCl_2_, 100 mM NaCl, 30 µM guanosine diphosphate (GDP)) in the presence of increasing concentrations of the tested compounds (10^−10^–10^−5^ M). Compounds were dissolved in ethanol, so that the concentration of ethanol in the assay did not exceed 0.75%. Non-specific binding was determined with 100 µM of unlabeled GTPγS. The reaction mixture was incubated for 90 min. at 37 °C in a volume of 250 µL. Next, 96-well Unifilter^®^ Plates (Perkin Elmer, Waltham, MA, USA) were presoaked for 1 h with 50 mM Tris-HCl (pH = 7.4) before harvesting. The reaction was terminated by vacuum filtration onto filter plates with the FilterMate Harvester^®^ (Perkin Elmer, Waltham, MA, USA). The samples were then rapidly washed with 2 mL of 50 mM Tris-HCl (pH = 7.4) buffer. Filter plates were dried for 2 h at 50 °C. After drying, 45 µL of EcoScint-20 scintillant (Perkin Elmer, Waltham, MA, USA) was added to every well. Radioactivity was counted in a Trilux MicroBeta^2^ counter (Perkin Elmer, Waltham, MA, USA). Data were analyzed with GraphPad Prism 5.0 software (GraphPad Software, San Diego, CA, USA). Curves were fitted with a one-site non-linear regression model. Efficacy (E_max_) and potency (EC_50_) were calculated from the Cheng-Prusoff equation and expressed as means ± SEM.

#### 4.2.2. 5-HT_2A_ Competition Binding Assay

Male Sprague-Dawley (SD) rats were decapitated, their brains removed and placed on ice. Frontal cortices were homogenized with a glass homogenizer in 30 vol. ice-cold homogenization buffer (50 mM Tris-HCl, 1 mM EDTA, 5 mM MgCl_2_, pH 7.4). Next, the homogenate was centrifuged at 20,000× *g* for 15 min at 4 °C. The pellet was suspended in 30 vol. 50 mM Tris-HCl (pH 7.4) and incubated in a water bath for 15 min. at 37 °C to remove endogenous serotonin. The suspension was again centrifuged at 20,000× *g* for 15 min at 4 °C. The pellet was resuspended in 10 vol. 50 mM Tris-HCl (pH 7.4) and the centrifugation step was repeated. The final pellet was suspended in 10 vol. 50 mM Tris-HCl (pH 7.4) and stored at −80 °C. For the 5-HT_2A_ assay frontal cortex homogenates (160 µg protein/mL) were incubated in triplicate with 1 nM [^3^H]ketanserin for 60 min. at 36 °C in a 50 mM Tris-HCl (pH 7.4) buffer containing 0.1% ascorbate, 3 mM CaCl_2_ and 10 µM pargyline) and increasing concentrations (10^−11^ M–10^−5^ M) of the compound of interest. Non-specific binding was determined in the presence of 10 μM mianserin. After incubation, the reaction mixture was deposited onto UniFilter-96 GF/B plates with the aid of a FilterMate-96 Harvester. Filter plates were presoaked beforehand with 0.4% PEI for 1 h. Next, each filter well was washed with 1.75 mL of 50 mM Tris-HCl (pH 7.4) and left to dry on a heating block set to 50 °C for 2 h. Then, 45 µL of Microscint-20 scintillation fluid was added to each filter well and left to equilibrate overnight. Filter-bound radioactivity was counted in a MicroBeta^2^ Microplate Counter. Binding curves were fitted with one site non-linear regression. Affinity was presented as the inhibitory constant (pKi and Ki ± SEM) from two or three separate experiments.

#### 4.2.3. 5-HT_2C_ Competition Binding Assay

The 5-HT_2C_ assay was performed in HEK 239 cells stably transfected with the human 5-HT_2C_ receptor (ChemiSCREEN™ Human 5-HT_2C_ Receptor Membrane Preparation, Merck). Membranes (5 μg/well final concentration) were mixed with 6 nM [^3^H]mesulergine in a 50 mM Tris-HCl (pH 7.4) buffer containing 5 mM MgCl_2,_ 1 mM CaCl_2_. Compounds were dissolved in 50% DMSO and added to the reaction mixture at 10 concentrations equally spaced on a log scale (10^−10^–10^−4.5^ M) and incubated for 2 h at 25 °C. The total DMSO concentration was 5%. Non-specific binding was determined in the presence of 10 μM mianserin. After incubation, the reaction mixture was deposited onto UniFilter-96 GF/B plates with the aid of a FilterMate-96 Harvester. Filter plates were presoaked beforehand with 0.33% polyethyleneimine (PEI) and 0.5% bovine serum albumin (BSA) for 30 min. Next, each filter well was washed with 1.75 mL of 50 mM Tris-HCl (pH 7.4) buffer supplemented with 500 mM NaCl to wash out the unbound ligands. Filters were and left to dry overnight at room temperature (RT). When completely dry, 45 µL of Microscint-20 scintillation fluid was added to each filter well and allowed to equilibrate for 30 min. Filter-bound radioactivity was counted in a MicroBeta^2^ Microplate Counter. Binding curves were fitted with a one site non-linear regression model. Affinity was calculated using the result of the Cheng–Prusoff equation (pKi ± SEM and Ki, 95% CI) from two separate experiments.

#### 4.2.4. Functional D_2_ Receptor Assay

Male Sprague-Dawley rats were decapitated, with their brains removed and placed on ice. Striatal tissue was dissected and homogenized with a glass homogenizer in 30 vol. ice-cold TED buffer (50 mM Tris-HCl, 1 mM EDTA, 1 mM dithiothreitol, pH 7.4). Next, the homogenate was centrifuged at 48,000× *g* for 15 min at 4 °C. The pellet was suspended in 30 vol. TED buffer (pH 7.4) and incubated in a water bath for 10 min. at 37 °C to remove endogenous ligands. The suspension was centrifuged again at 48,000× *g* for 15 min at 4 °C. The pellet was resuspended in 30 vol. TED buffer (pH 7.4) and the centrifugation step was repeated. The final pellet was suspended in 10 vol. 50 mM Tris-HCl (pH 7.4) and stored at −80 °C until use. For the D_2_ receptor antagonist [^35^S]GTPγS assay, 15μg/mL of striatal homogenate was incubated in triplicate with 0.8 nM [^35^S]GTPγS (final concentration: 0.08 nM) in an assay buffer (50 mM Tris-HCl, pH = 7.4, 1 mM EGTA, 3 mM MgCl_2_, 100 mM NaCl, 0.1 mM dithiothreitol, 500 µM ascorbic acid, 20 µM GDP and 40 µM dopamine) in the presence of increasing concentrations of the tested compounds (10^−8^−10^−3.5^ M). Serial dilutions of compounds were dissolved in 7.5% ethanol. Dopamine was dissolved in 50 mM Tris buffer (pH = 7.4) supplemented with 500 µM ascorbic acid to prevent oxidation. The effect on basal G-protein activation threshold was determined in assay buffer deprived of dopamine. The final ethanol concentration in the assay was 0.75%. Non-specific binding was determined with 100 µM of unlabeled GTPγS. The reaction mixture was incubated for 60 min. at 30 °C at a volume of 250 µL. Next, 96-well Unifilter^®^ Plates (Perkin Elmer, Waltham, MA, USA) were presoaked for 1 h with 50 mM Tris-HCl (pH = 7.4) before harvesting. The reaction was terminated by vacuum filtration onto filter plates with the FilterMate Harvester^®^ (Perkin Elmer, Waltham, MA, USA). Samples were then rapidly washed with 2 mL of 50 mM Tris-HCl (pH = 7.4) buffer. Filter plates were dried overnight at RT. After drying, 45 µL of EcoScint-20 scintillant (Perkin Elmer, Waltham, MA, USA) was added to the wells. Radioactivity was counted in a Trilux MicroBeta^2^ counter (Perkin Elmer, Waltham, MA, USA). Data were analyzed with GraphPad Prism 5.0 software (GraphPad Software, San Diego, CA, USA). Curves were fitted with a one-site non-linear regression model. Inhibitory potency (pIC_50_) was expressed as means ± SEM. IC_50_ values were expressed in µM and presented as means ±95% CI from two separate experiments.

### 4.3. Computational Methods

#### 4.3.1. Compound Preparation

The investigated compounds **Norbo1-28** were modeled using LigPrep module [47] of Schrödinger suite of software, v. 2019-4 as previously reported [22,23]. In order to find the protonation state, Epik module [48] of Schrödinger suite of software, v. 2019-4 was used.

#### 4.3.2. Receptor Structures

X-ray structures of respective receptors in inactive conformations were used: serotonin 5-HT_2A_ receptor in complex with an antagonist risperidone (PDB ID: 6A93 [49]) and serotonin 5-HT_2C_ receptor in complex with an inverse agonist ritanserin (PDB ID: 6BQH [50]) as previously reported [22,23]. In the case of serotonin 5-HT_1A_ receptor, the cro-EM structure of the receptor in active conformation in complex with a partial agonist aripiprazole (PDB ID: 7E2Z [51]) was used. The structures of the receptors were preprocessed using the Protein Preparation Wizard of Maestro Release 2019.4 [52] as previously reported [22,23]. The Yasara Structure v. 20.12.24 [53] tool for loop modelling was used to build receptors extracellular loops if necessary.

#### 4.3.3. Molecular Docking

Standard Precision (SP) method of molecular docking with Glide [54] from Schrödinger release 2019-4 was applied. The grid files were obtained based on co-crystallized ligands. The selected receptors hydroxyl groups in the active sites were made flexible as previously reported [22,23]. Hence, 200 poses were generated in the case of each ligand-receptor complex. The poses were then filtered to ensure the interaction of the conserved Asp3.32 (Ballesteros-Weinstein numbering [55]) of the receptor with a protonatable nitrogen atom of the ligand. The final poses were identified by consideration of Glide docking scores and based on the visual inspection. Maestro Release 2019.4 [52] and PyMol 2.0.4 [56] software were used for visualization of molecular modeling results.

### 4.4. In Vivo Behavioral Tests

#### 4.4.1. General Procedures

The experiments were carried out, in the Experimental Medicine Center of Medical University of Lublin, on six-week-old naive male Swiss mice, weighing 24–30 g. The mice were housed in cages, five individuals per cage in an environmentally controlled rooms (ambient temperature 22  ±  1 °C; relative humidity 50–60%; 12 h light/dark cycle, lights on at 8:00). Standard laboratory food(LSM-Agropol-Motycz, Lublin, Poland) and filtered water were available ad libitum. All the experimental procedures were carried out, according to the National Institute of Health Guidelines for the Care and Use of Laboratory Animals and to the European Community Council Directive for the Care and Use of Laboratory Animals of 22 September 2010 (2010/63/EU) and approved by the Local Ethics Committee for Animal Experimentation in Lublin. The investigated compounds (**Norbo-4**, **Norbo-8**, **Norbo-10**, **Norbo-14**, **Norbo-18** and **Norbo-20**) in all tests were administered intraperitoneally (i.p.), dissolved in dimethyl sulfoxide (DMSO, final concentration of 0.1%) and then diluted by aqueous solution of 0.5% methylcellulose (tylose) and injected 60 min before the tests. Other drugs (amphetamine (amph), buspirone, sertraline) were administered i.p. diluted in saline, and injected30 (amph) or 60 min (buspirone, sertraline) before the tests. All compounds were given in a volume of 10 mL/kg to mice. The control animals received an equivalent volume of the solvent at the respective time before the tests. All the experiments were conducted in the light phase between 09.00 a.m. and 14.00 p.m. The experiments were performed by an observer unaware of the treatment administered.

#### 4.4.2. Spontaneous Locomotor Activity

The locomotor activity of mice was measured using an animal activity meter Opto-Varimex-4 Auto-Track (Columbus Instruments, Columbus, OH, USA). This automatic device consists of eight transparent cages with a lid, set of four infrared emitters (each emitter has 16 laser beams) and eight detectors monitoring animal movements. To assess the spontaneous activity of mice, the compounds or vehicle (as a control) were administered 60 min before the test. In another set of experiments, the influence of tested compounds on amph-induced hyperactivity in mice was evaluated. The study was conducted in the same apparatus, but each mouse received amph (5 mg/kg, s. c.) 30 min after injection of vehicle or tested compounds. The animals were placed in the cage individually, 50 min after the administration of the tested compounds, for a period of 10 min for acclimatization. After this time, their activity was noted after 6 min (corresponded with the time duration of the FST and near (5 min) to the EPM test, respectively) and after 20 min, to observe the dynamics of changes. The distance travelled in centimeters [cm] was measured. The cages were cleaned up with 10% ethanol after each mouse.

#### 4.4.3. Motor Coordination

The effects of investigated compounds on motor coordination were evaluated in the rota-rod [57] and chimney [58] tests. In the first test, motor impairments were measured, defined as the inability to keep balance on a rotating rod (at constant speed of 18 rpm) for 1 min. In the second test, motor impairments were assessed by mouse inability to climb up the tube backwards (3 cm in inner diameter, 25 cm long) within 60 s. Before the tests, the animals were trained once a day for 3 days. The animals able to stay on the rotating rod or to leave the chimney for 60 s were approved for experiments.

#### 4.4.4. FST (Porsolt’s Test) in Mice

The experiment was carried out according to the method of Porsolt et al. [59]. Mice were individually placed in a glass cylinder (25 cm high; 10 cm in diameter) containing water maintained at 23–25 °C, and were left there for 6 min. The total duration of immobility was recorded during the last 4 min of a 6-min test session. A mouse was regarded as immobile when it remained floating on the water, making only small movements to keep its head above the water.

#### 4.4.5. EPM Test

The EPM studies were carried out on mice according to the method of Lister [60]. The EPM apparatus was made of plexiglass and consisted of two open (30 × 5 cm) and two enclosed (30 × 5 × 15 cm) arms. The arms extended from a central platform of 5 × 5 cm. The apparatus was mounted on a plexiglass base, raising it 38.5 cm above the floor, and illuminated by a red light. The test consisted of placing a mouse in the center of the apparatus (facing an open arm) and allowing it to freely explore. The number of entries into the open arms and the time spent in these arms were scored for a 5 min test period. An entry was defined as placing all four paws within the boundaries of the arm. The following measures were obtained from the test: the total number of arm entries; the percentage of arm entries into the open arms; and the time spent in the open arms expressed as a percentage of the time spent in both the open and closed arms. Anxiolytic activity was indicated by increases in the time spent in open arms and in the number of open arm entries. The total number of entries into either type of arm was used additionally as a measure of overall motor activity.

#### 4.4.6. Statistical Analysis

The results were calculated by the one-way analysis of variance ANOVA, followed by Dunnett’s post-hoc test. The results are presented as mean ± standard errors (SEM). The level of *p* < 0.05 was considered as statistically significant. All the figures were prepared by the GraphPad Prism version 5.00 for Windows, GraphPad Software (San Diego, CA, USA).

### 4.5. Ex Vivo Assays

#### 4.5.1. General Procedures

Male rats (Sprague-Dawley, 160–200 g; Harlan Laboratories, S. Pietro al Natisone, Italy) were manipulated and cared for in strict compliance with the principles of laboratory animal care (NIH publication n° 86–23, revised 1985) and the Italian D.L. no. 116 of 27 January 1992 and associated guidelines in the European Communities Council Directive of 24 November 1986 (86/609/ECC). Animal housing complied with recent pharmacological guidance [61]. All animals weighing 160–200 g were used after a 1-week acclimation period (temperature 23 ± 2 °C; humidity 60%, free access to water and standard food).

#### 4.5.2. Ileum Preparation and Evaluation of 5-HT-Evoked Contractions

Rats were asphyxiated using CO_2_ and segments (1–1.5 cm) of ileum were removed, flushed of luminal contents, and placed in Krebs solution (119 mM NaCl, 4.75 mM KCl, 1.2 mM KH_2_PO_4_, 25 mM NaHCO_3_, 2.5 mM CaCl_2_, 1.5 mM MgSO_4_, and 11 mM Glucose). The segments were prepared as previously described [62]: the segments were set up in such a way as to record contractions mainly from the longitudinal axis, in an organ bath containing 20 mL of Krebs solution, bubbled with 95% O_2_ and 5% CO_2_ and maintained at 37 °C. The tissues were connected to an isotonic transducer (load: 0.5 g), connected to PowerLab system (Ugo Basile, Comerio, Italy). Ileal segments were equilibrated for 60 min [63] followed by three repeated additions of submaximal concentration of 5-HT (10^−5^ M) in order to record stable control contractions. To evaluate the inhibitory activity, the responses were observed in the presence of increasing concentrations (10^−8^–10^−5^ M). In preliminary experiments, the effect of 5-HT was observed in the presence of the neuronal blocker tetrodotoxin (0.3 µM), the muscarinic receptor antagonist atropine (1 µM), the adrenergic receptor antagonists phentolamine (10^−6^ M) plus propranolol (10^−6^ M) and the 5-HT_2A_ antagonist ketanserin (0.1 µM). The contact time for each concentration was 10 min. The compounds were dissolved in DMSO. DMSO (<0.01%) did not modify 5-HT-induced contractions. Results are expressed as mean (SEM). The concentration of the compounds that produced 50% inhibition of 5-HT-induced contractions (IC_50_) or maximal inhibitory effect (E_max_) were used to characterize compounds potency and efficacy, respectively. The IC_50_ and E_max_ values were calculated with the aid of a computer program (Graphpad Prism 5).

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
