# Peer review of "Synthesis, Docking Studies and Pharmacological Evaluation of Serotoninergic Ligands Containing a 5-Norbornene-2-Carboxamide Nucleus"

_molecules, 2022, doi:10.3390/molecules27196492_

Round 1
Reviewer 1 Report
Authors have done a good experimental and overall all manuscript is well written so it can be published in the present form.
Author Response
I want to thank the reviewer for the favorable response
Reviewer 2 Report
g
Manuscript entitled “Synthesis, Docking studies and Pharmacological Evaluation of Serotoninergic Ligands containing a 5-norbornene-2-carboxamide nucleus” submitted by Sparaco, R et al. describes the synthesis of class of a 5-norbornene-2-carboxamide derivatives for the pharmacologic evaluation of serotoninergic receptors. Please see comments below:
11) In scheme 1, the authors have shown two step synthesis sequence where the second step yielded poor yields (30-40%), Did the authors explored other conditions to improve the yields? Please explain.
22) Table 2 and Table 3 were labeled or placed wrong. Please correct.
33) In table 3, how does the authors compare pIC50 and IC50 of Norbo 1-28 with domperidone? Does the derivatives shwows significant antagonistic activity?
44) In table 2 of current version, How does the Ki values of Norbo 1-28 for 5-HT2a and 5-HT2c were compared with respect to ketanserin, seronin and RS102221 respectively. Please explain the rationale.
55) When the authors docked Norbo 1-28 to the orthosteic sites of 5-HT1a, 5-HT2a and 5-HT2c receptors, why did the authors didn’t observe same or similar efficacy order as they observed in in vitro evaluations?
66) For fig 5A and 5B, please include y-axis data for different dosages for Norbo 8, 10 and 20 for fair comparison and discuss the differences.
77) For fig 6, please include the data for different dosages of Norbo -10 and 20 and discuss the results.
88) Overall, the manuscript entails poor synthesis and not much of a considerable pharmacological activity of the compounds synthesized.
Major revisions needs to be addressed.
Author Response
- In scheme 1, the authors have shown two step synthesis sequence where the second step yielded poor yields (30-40%), Did the authors explored other conditions to improve the yields? Please explain.
- We performed some different synthetic procedures without increasing reaction yields
- Table 2 and Table 3 were labeled or placed wrong. Please correct.
- We corrected in the text
- In table 3, how does the authors compare pIC50 and IC50 of Norbo 1-28 with domperidone? Does the derivatives shwows significant antagonistic activity?
- Inhibitor potencies of compounds relative to domperidone were calculated by means of one-way ANOVA followed by the Bonferroni post-hoc analysis. A suitable table legend was added to explain the statistics of those comparisons. Overall, the compounds tested showed minute D2 antagonist properties that most likely have little to no pharmacological significance. This is why in the original version we did not indicate differences in potency related to the 4-phenylpiperazin-1-yl substituted isomer. However, to avoid inconsistencies, we decided to include those comparisons as well. In general, for more clarity we chose not to indicate differences when a compound does not show superior potency/efficacy/affinity compared with the reference compound.
- In table 2 of current version, How does the Ki values of Norbo 1-28 for 5-HT2a and 5-HT2c were compared with respect to ketanserin, seronin and RS102221 respectively. Please explain the rationale.
- In Table 2 potencies were statistically analyzed with one-way ANOVA and the Bonferroni post-hoc test. 5-HT2A and 5-HT2C affinities were compared with affinities of selective ligands - ketanserin and RS102221, respectively as well as non-selective serotonin. The rationale behind comparing the query compounds to those reference compounds, stems from the fact that the goal of many drug design programmes is to search for compounds that surpass the affinities of top known selective ligands at the receptor in question. For that a reference compound is essential. Thus, comparisons with high-affinity reference compounds with documented pharmacological activity in part predicts the possible pharmacological action of the query compound. Moreover, high affinity translates to higher receptor occupancy and potency, which translates to lower side-effect liability associated with dose elevation in order to achieve high receptor occupancy.
- When the authors docked Norbo 1-28 to the orthosteic sites of 5-HT1a, 5-HT2a and 5-HT2c receptors, why did the authors didn’t observe same or similar efficacy order as they observed in in vitro evaluations?
-
To the best of our knowledge it is not correct to correlate a docking score with ligand efficacy. The term ligand efficacy refers to the capability of a molecule to induce a specific physiological response of receptor activation (Kim et al. 2020) and it is not only affected by the binding energy. In contrast, docking score reflects to some extent binding affinity. However, the accuracy of binding affinity prediction from the docking score is rather low and it is not a recommended practice to correlate it with the experimental affinity values. In our manuscript we report Ki values for Norbo 1-28 and 5-HT2A and 5-HT2c receptors. The compounds for which molecular docking results are shown have similar Ki values – from 13 to 51 nM. This is the range of Ki which is not recognized as differentiated in the docking score so, in spite of different number of hydrogen bonds formed by the ligands in particular in case of 5-HT2C receptor, the obtained Glide docking scores are comparable for all the ligands shown.
Kim M, Wei J-D, Harmalkar DS, Goo J-I, Lee K, Choi Y, Kim J-H, Cho AE. 2020. Elucidation of Mechanism for Ligand Efficacy at Leukotriene B4 Receptor 2 (BLT2). ACS Med Chem Lett. 11(8):1529–1534. https://doi.org/10.1021/acsmedchemlett.0c00065
- For fig 5A and 5B, please include y-axis data for different dosages for Norbo 8, 10 and 20 for fair comparison and discuss the differences.
- For fig 6, please include the data for different dosages of Norbo -10 and 20 and discuss the results.
-
We thank the Referee for the suggestions. We discontinued the measurements for the subsequent doses of Norbo 8, 10 and 20 (Figure 5A and 5B) and Norbo 10 and 20 (Figure 6) since the data obtained in experiments and presented on the above graphs clearly indicated no significant effect of these doses. Our choice was dictated by the desire to limit the number of animals used in experiments in accordance with the requirements of the 3R principle. In accordance with this principle, the limited number of animals allowing for obtaining reliable behavioral results and enabling the possibility of statistical analysis was planned in all experiments.
Reviewer 3 Report
Page 1, lines 39-40: neurotransmitter systems is not the right term. Serotonin is not a system; it is a neurotransmitter or a chemical mediator.
The acronyms CNS, LCAPs, 8-OH-DPAT, GPCRs, FST, EPM, SERT, GDP, etc. must be explained at the first appearance in the text.The authors are asked to explain why they did not use a reference substance for the in vivo tests: motor coordination and spontaneous locomotor activity?
In chemical names, the prefix exo instead of eso.
The authors are asked what is the reason why the compounds Norbo-19, 25, 27, 28 were transformed into salts with hydrochloric acid and if the obtained salts were characterized physico-chemically.
Norbo-25 compound has a very wide melting range. Is it the hydrochloric acid salt?
Page 32, line 798: probably EDTA instead of EGTA.
Author Response
Page 1, lines 39-40: neurotransmitter systems is not the right term. Serotonin is not a system; it is a neurotransmitter or a chemical mediator.
We corrected in the text
The acronyms CNS, LCAPs, 8-OH-DPAT, GPCRs, FST, EPM, SERT, GDP, etc. must be explained at the first appearance in the text.
We appropriately corrected in the text
The authors are asked to explain why they did not use a reference substance for the in vivo tests: motor coordination and spontaneous locomotor activity?
In both tests we used a 0.9% NaCl (saline) as a reference substance. This is in line with the procedures used by us and other authors in behavioral research when we want to show that compounds do not affect the behavior of animals. 0.9% NaCl was used because it does not affect the behavior of animals and its injection is only aimed at maintaining conditions comparable to other groups - injection stress and administration of the solution.
In chemical names, the prefix exo instead of eso.
We corrected in the text
The authors are asked what is the reason why the compounds Norbo-19, 25, 27, 28 were transformed into salts with hydrochloric acid and if the obtained salts were characterized physico-chemically.
Some compounds were converted into the hydrochloride salts because they were liquid at room temperature and this made biological assays difficult. However, for these compounds the characterization was performed on the hydrochloride salts
Norbo-25 compound has a very wide melting range. Is it the hydrochloric acid salt?
We corrected in the text a typing error
Page 32, line 798: probably EDTA instead of EGTA.
It is not an error because EGTA was used in the experiment and the acronym was specified in the text
Round 2
Reviewer 2 Report
Please accept the current version